# Cycle-Sync: Robust Global Camera Pose Estimation through Enhanced Cycle-Consistent Synchronization

**Shaohan Li**[*]        **Yunpeng Shi**[†]        **Gilad Lerman**[*]

[*]School of Mathematics, University of Minnesota
[†]Department of Mathematics, University of California, Davis
nicklsh1996@gmail.com, lerman@umn.edu, ypshi@ucdavis.edu

## Abstract

We introduce Cycle-Sync, a robust and global framework for estimating camera poses (both rotations and locations). Our core innovation is a location solver that adapts message-passing least squares (MPLS)—originally developed for group synchronization—to camera location estimation. We modify MPLS to emphasize cycle-consistent information, redefine cycle consistencies using estimated distances from previous iterations, and incorporate a Welsch-type robust loss. We establish the strongest known deterministic exact-recovery guarantee for camera location estimation, showing that cycle consistency alone—without access to inter-camera distances—suffices to achieve the lowest sample complexity currently known. To further enhance robustness, we introduce a plug-and-play outlier rejection module inspired by robust subspace recovery, and we fully integrate cycle consistency into MPLS for rotation synchronization. Our global approach avoids the need for bundle adjustment. Experiments on synthetic and real datasets show that Cycle-Sync consistently outperforms leading pose estimators, including full structure-from-motion pipelines with bundle adjustment.

## 1   Introduction

Structure-from-Motion (SfM) is a central task in 3D computer vision [21], aimed at reconstructing the 3D structure of a scene from 2D images captured by cameras with unknown poses. It plays a critical role in applications such as virtual and augmented reality, robotics, and autonomous driving. A core challenge in SfM is accurate camera pose estimation, which is also foundational to modern 3D techniques such as neural radiance fields [17] and Gaussian splatting [8], where camera parameters serve as priors or inputs for rendering and synthesis.

A typical SfM pipeline begins by estimating essential matrices between pairs of cameras, from which local pose information is extracted. It then infers absolute camera orientations from relative ones—a step commonly referred to as rotation synchronization (or averaging). Next, it estimates camera locations from relative direction vectors. Once the camera poses are determined, a standard final step is to recover the 3D structure of the scene. The aim of this work is to revisit both pose estimation tasks and propose global solutions that eliminate reliance on the highly incremental and computationally intensive bundle adjustment step [31], which is ubiquitous in current SfM pipelines. We refer to our robust location solver as *Cycle-Sync*[1], and use the same name for the full pipeline that incorporates this solver along with robust direction and rotation estimation.

Mathematically, these problems are formulated on a graph $G([n], E)$, where nodes correspond to cameras, and an edge $(i, j) \in E$ indicates the availability of relative pose information between cameras $i$ and $j$. The goal of rotation synchronization is to estimate the ground-truth camera rotations

---

[1]The Matlab code is released at `https://github.com/sli743/Cycle-Sync`

$\{\boldsymbol{R}_i^*\}_{i\in[n]}$ from noisy relative rotations $\{\boldsymbol{R}_{ij}\}_{ij\in E}$, whose clean counterparts satisfy $\boldsymbol{R}_{ij}^* = \boldsymbol{R}_i^* \boldsymbol{R}_j^{*\top}$. This recovery is up to an arbitrary global rotation. The subsequent localization step seeks to estimate the absolute positions $\{\boldsymbol{t}_i^*\}_{i=1}^n$ from noisy unit direction vectors $\{\boldsymbol{\gamma}_{ij}\}_{ij\in E}$, which approximate the ground-truth directions:

$$\boldsymbol{\gamma}_{ij}^* := (\boldsymbol{t}_i^* - \boldsymbol{t}_j^*)/\|\boldsymbol{t}_i^* - \boldsymbol{t}_j^*\|.$$

These locations are only recoverable up to a global translation and scale.

Camera location estimation is significantly more challenging than rotation estimation, and is therefore the primary focus of our work. First, the direction vectors derived from essential matrices lack scale information. If scale were available, the problem would reduce to a special case of group synchronization over the translation group, making it considerably easier. Second, in modern SfM pipelines, the estimated direction vectors are often highly corrupted due to failures in feature matching or RANSAC, as well as error propagation from earlier stages—particularly from rotation estimates. Lastly, unlike rotations, the space of camera locations lacks a group structure and is non-compact, which makes location estimation more sensitive and numerically unstable in the presence of corruption and noise.

## 1.1 Relevant previous works

Since our main contributions focus on camera location estimation, we review the most relevant work on this task, while also briefly noting related advances in rotation synchronization. Early location solvers based on $\ell_2$ minimization [1, 2, 6, 20, 32, 33] or $\ell_\infty$ minimization [18] are highly sensitive to outliers and unsuitable for real-world Structure-from-Motion (SfM) data. More robust approaches, including Least Unsquared Deviations (LUD) [19] and ShapeFit [7, 5], use convex $\ell_1$ objectives, solved via Iteratively Reweighted Least Squares (IRLS) and the Alternating Direction Method of Multipliers (ADMM), respectively. These methods minimize the distance between $\boldsymbol{t}_i - \boldsymbol{t}_j$ and the line defined by $\boldsymbol{\gamma}_{ij}$, which can overweight long edges and become unstable when edge lengths vary significantly or are corrupted. BATA [36] instead minimizes the sine of the angle between $\boldsymbol{t}_i - \boldsymbol{t}_j$ and $\boldsymbol{\gamma}_{ij}$, offering robustness to edge-length variation. However, it treats all edges equally and thus underutilizes information from clean long edges. Fused Translation Averaging (fused-TA) [15] alternates between LUD- and BATA-type objectives and merges their outputs using uncertainty estimates, but may underperform both in practice, with no clear winner among the three. Both 1-Dimensional SfM (1DSfM) [34] and All-About-that-Base (AAB) [25] exploit 3-cycle consistency to detect outliers. 1DSfM projects directions to 1D and applies a heuristic combinatorial algorithm, without theoretical guarantees. AAB enforces coplanarity of $\boldsymbol{\gamma}_{ij}^*$, $\boldsymbol{\gamma}_{jk}^*$, and $\boldsymbol{\gamma}_{ki}^*$ and combines this constraint with message passing. AAB outperforms 1DSfM empirically but is unstable in near-colinear configurations and only achieves approximate guarantees under a specific probabilistic model.

The only deterministic recovery guarantees for location estimation under adversarial corruption are those for ShapeFit [7] and LUD [12], under Gaussian location priors and Erdős–Rényi measurement graphs with number of nodes approaching infinity and certain bounds on the connection probability and the number of corrupted incident edges per node.

In the related problem of group synchronization, stronger guarantees exist. Cycle-Edge Message Passing (CEMP) [11] handles adversarial and probabilistic corruption in rotation synchronization. It was used to create Message-Passing Least Squares (MPLS) [26], which empirically improves performance on camera orientation estimation. However, MPLS progressively downweights cycle information and cannot be extended to location estimation due to the lack of inter-camera distance measurements. Similarly inspired by CEMP, DESC [28] estimates edge corruption levels via quadratic programming, but it is computationally slow and its theoretical guarantees do not cover adversarial corruption. DDS [16] is able to address adversarial corruption by exploiting Tukey depth in the tangent space of $SO(d)$. Other extensions attempt to generalize CEMP or MPLS to permutation synchronization [27] and partial permutation synchronization [14], yet these methods and their theory also do not naturally extend to the location estimation problem.

Finally, several global SfM pipelines incorporate location solvers. These include LUD [19] (used as a full pipeline), Theia [30], and GLOMAP [22]. All of them benefit from non-global bundle adjustment [31], which is explicitly integrated into Theia and GLOMAP.

## 1.2 Contributions

Nearly all existing methods for location estimation fall within the IRLS framework and do not fully exploit cycle consistency information. As a result, they struggle to handle cycle-consistent corruption (i.e., situations when the corrupted edges exhibit cycle-consistent behavior), which frequently arises in real-world SfM datasets.

The goal of this work is to propose a new location estimation method that is robust to severe corruption and accommodates missing and highly variable edge lengths. We also revisit the global pipeline for pose estimation and improve several of its core components.

Our contributions to camera location estimation are summarized as follows:

1. We introduce a novel formulation with a Welsch-type objective function that directly addresses key limitations of LUD-type and BATA-type objectives, particularly in the presence of large variations in edge distances.

2. We propose a new MPLS framework to optimize the Welsch objective for location estimation. This framework is designed to fully exploit cycle-consistent information. To handle missing distance data, we redefine cycle consistencies using distances estimated in previous iterations.

3. We establish the strongest known deterministic exact-recovery guarantee for location estimation under adversarial corruption. Under standard probabilistic models (e.g., i.i.d. Gaussian locations with Erdős–Rényi connectivity), our theoretical sample complexity improves over all prior work (see Table 1).

Our additional contributions to global pose estimation include:

1. We extend the full-cycle MPLS framework to the rotation synchronization problem, yielding significantly reduced orientation error compared to the strongest existing baselines.

2. We introduce a plug-and-play outlier rejection module inspired by robust subspace recovery. This module significantly improves the performance of existing location estimators.

3. Our global pose estimation pipeline eliminates the need for bundle adjustment. Experiments on both synthetic and real datasets show that Cycle-Sync consistently outperforms leading pose estimators, including full structure-from-motion pipelines that rely on bundle adjustment.

## 2 The Cycle-Sync Framework

We first present the Cycle-Sync location solver: §2.1 introduces its optimization formulation, §2.2 details its algorithmic implementation with cycle-consistent weighting, and §2.3 establishes theoretical recovery guarantees. Finally, §2.4 integrates the solver into a full camera pose estimation pipeline.

### 2.1 New optimization formulation for location estimation

We note that all major solutions to the camera location estimation problem are either special cases or variants of the following formulation:

$$\min_{\{t_i\}_{i=1}^n, \{\alpha_{ij}\}_{ij \in E}} \sum_{ij \in E} \rho(\|t_i - t_j - \alpha_{ij}\gamma_{ij}\|) \tag{1}$$

$$\text{subject to} \quad \alpha_{ij} \geq 1, \quad \sum_i t_i = 0,$$

where $\rho(x)$ is a certain function to be specified later. Here, $\alpha_{ij}$ is interpreted as an auxiliary variable representing the distance between $t_i$ and $t_j$. Its constraint, i.e., $\alpha_{ij} \geq 1$, aims to avoid the trivial solution $t_i = 0$ for all $i$'s. This general formulation captures several existing methods as special cases. For example, constrained least squares [32, 33] uses $\rho(x) = x^2$ and LUD [19] uses $\rho(x) = |x|$. Furthermore, ShapeFit [7] replaces $\|t_i - t_j - \alpha_{ij}\gamma_{ij}\|$ by the projection distance from $t_i - t_j$ to the line of $\gamma_{ij}$. It also switches the $\alpha_{ij} \geq 1$ to a weaker linear constraint $\sum_{ij \in E} \langle t_i - t_j, \gamma_{ij} \rangle = 1$. Let $\theta_{ij}$ denote the angle between the true relative location $t_i - t_j$ and direction $\gamma_{ij}$. By choosing the optimal $\alpha_{ij}$, the LUD objective function (with $\rho(x) = |x|$) reduces to the ShapeFit one:

$$\rho(\|t_i - t_j - \alpha_{ij}^*\gamma_{ij}\|) = \rho(\|t_i - t_j\| \sin \theta_{ij}),$$

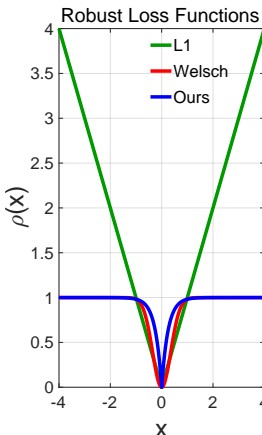

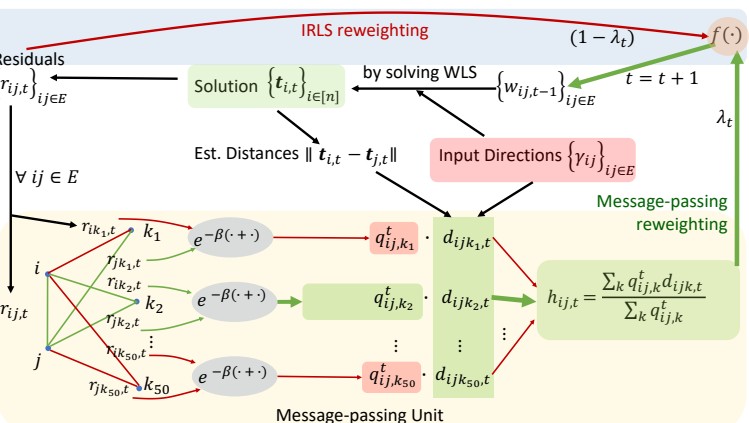

Figure 1: Comparison of different losses. Our loss combines the advantages of the $L_1$ and Welsch losses. Specifically, it suppresses the influence of large $x$ values (like Welsch), while retaining a nonsmooth corner at the origin (like $L_1$), which introduces a singular point in the reweighting function $f(x)$ and enables exact recovery.

Figure 2: Illustration of Cycle-Sync. The algorithm solves the weighted least squares (WLS) in (3) to estimate the locations $\{t_i\}_{i \in [n]}$. The weights for WLS are iteratively updated in two ways. The main one is a cycle-edge message passing procedure (bottom unit, where green elements represent cleaner information, and red elements indicate corrupted information). Each cycle weight $q_{ij,k}^t$ is updated using the two residuals $r_{ik,t}$ and $r_{jk,t}$. The bar length surrounding $q_{ij,k}^t$ reflects its magnitude. The theory indicates higher weights for good (green) cycles. The quantity $h_{ij,t}$ is a weighted average of $d_{ijk,t}$, defined in (7)) and updated at each iteration using the estimated locations. The theory guarantees that $h_{ij,t}$ is a good estimate for the corruption level at edge $ij$. The weights are obtained by applying the function $f(x)$ in (4). The weights are also computed by IRLS (top unit). The final weights combine the two procedures with $\lambda_t \to 1$, so the message-passing unit progressively takes over.

which scales linearly with the length of edges. Therefore, such an objective function may suffer from long and corrupted edges. BATA [36] changes the distance to $\rho(\sin \theta_{ij})$ in order to make it independent of $\|t_i - t_j\|$. However, it may not sufficiently benefit from long and clean edges that have a stronger effect on the global distribution of the locations.

As opposed to previous robust formulation which used $\rho(x) = |x|$, we propose minimizing (1) with

$$\rho(x) = 1 - e^{-a|x|}, \tag{2}$$

where $a$ is a fixed parameter. Throughout this paper we fix $a = 4$. We remark that this $\rho(x)$ is similar to the Welsch objective function $\rho(x) = 1 - e^{-ax^2}$. The latter objective functions put less emphasis on longer edges, but still grow (with a very small rate) as $\|t_i - t_j\|$ grows. Therefore, they are much less sensitive to large variations of distances. This avoids the limitations of both LUD and BATA. A comparison of these $\rho(x)$ choices appears in Figure 1.

On the other hand, unlike the traditional Welsch function, our new energy function inherits from LUD the non-smoothness at $x = 0$. This non-differentiability of $\rho(x)$ at 0 makes the exact recovery of ground truth locations possible under corrupted directions [12].

## 2.2 Solution of the proposed optimization with emphasis on cycles

The formulation proposed in (1) and (2) can be addressed by an IRLS-type approach. Indeed, let $r_{ij} := \|t_i - t_j - \alpha_{ij}\gamma_{ij}\|$, then our objective function is $F(\{t_i\}_{i=1}^n, \{\alpha_{ij}\}_{ij \in E}) := \sum_{ij \in E} \rho(r_{ij})$. As demonstrated in [3], the problem can be tackled with an IRLS scheme:

$$\min_{\{t_i\}_{i=1}^n, \{\alpha_{ij}\}_{ij \in E}} \sum_{ij \in E} w_{ij,t} \|t_i - t_j - \alpha_{ij}\gamma_{ij}\|^2. \tag{3}$$

Typically, $w_{ij,t+1}$ is updated by $f(r_{ij,t})$, where residual $r_{ij,t} = \|\boldsymbol{t}_{i,t} - \boldsymbol{t}_{j,t} - \alpha_{ij,t}\boldsymbol{\gamma}_{ij}\|$, and the reweighting function

$$f(x) = \rho'(x)/(x + \delta). \tag{4}$$

Here $\delta$ is a small number to avoid the zero denominator. However, our choice of $\rho(x)$ in (2) is nonconvex, making IRLS highly sensitive to weight initialization. Even with good initialization and convex $\rho$, IRLS may fail to recover the ground truth under high corruption. Indeed, in the highly corrupted scenario, the residuals $r_{ij,t}$ may fail to reflect true corruption levels (defined up to an unknown fixed scale):

$$s_{ij}^* = \|\boldsymbol{t}_i^* - \boldsymbol{t}_j^*\| \cdot \|\boldsymbol{\gamma}_{ij} - \boldsymbol{\gamma}_{ij}^*\|, \tag{5}$$

making IRLS easily get stuck in local minima. This limitation motivates a much more robust iterative estimator for $s_{ij}^*$, using the cycle-consistency information. Following the MPLS strategy [26] we approximate $s_{ij}^*$ using a weighted average of cycle inconsistencies:

$$s_{ij,t} = \frac{1}{Z_{ij,t}} \sum_{k \in N_{ij}} e^{-\beta(r_{ik,t}+r_{jk,t})} d_{ijk,t}, \tag{6}$$

where $\beta > 0$ is a parameter, $Z_{ij,t}$ is the normalization factor ensuring convex combinations, and

$$d_{ijk,t} = \left\| \|\boldsymbol{t}_{i,t} - \boldsymbol{t}_{j,t}\|\boldsymbol{\gamma}_{ij} + \|\boldsymbol{t}_{j,t} - \boldsymbol{t}_{k,t}\|\boldsymbol{\gamma}_{jk} + \|\boldsymbol{t}_{k,t} - \boldsymbol{t}_{i,t}\|\boldsymbol{\gamma}_{ki} \right\| \tag{7}$$

measuring the cycle inconsistency. We note that with clean edges and true locations, $d_{ijk,t} = 0$. Furthermore, as $\beta \to \infty$ and $\boldsymbol{t}_{i,t} \to \boldsymbol{t}_i^*$, $s_{ij,t} \to s_{ij}^*$.

We note that in (6), the corruption level is no longer approximated by a single residual. Instead, it aggregates information from 3-cycles (indexed by $ijk$) and incorporates residuals from neighboring edges, making $s_{ij,t}$ significantly more stable and robust to outliers.

In early iterations, when distance estimates $\|\boldsymbol{t}_{i,t} - \boldsymbol{t}_{j,t}\|$ are unreliable, we blend the residuals $r_{ij,t}$ with the corruption scores $s_{ij,t}$ to define the edge weights. As estimates improve, the weighting gradually shifts toward $s_{ij,t}$. This message-passing mechanism enables cycle consistency to refine edge weights, while the weighted least squares updates, in turn, refine cycle inconsistencies. This bi-directional communication between cycles and edges facilitates global information propagation and significantly reduces the risk of getting trapped in local minima. We refer to our method as Cycle-Sync, and its information propagation is illustrated in Figure 2.

**Weight initialization.** In the first iteration, when locations are unknown, weights can be initialized as $w_{ij,0} = \exp(-20\,\tilde{s}_{ij})$, where $\tilde{s}_{ij}$ estimates the angular corruption level. Its ground-truth counterpart is defined as

$$\tilde{s}_{ij}^* := \frac{1}{\pi}\angle(\boldsymbol{\gamma}_{ij}, \boldsymbol{\gamma}_{ij}^*) \in [0, 1]. \tag{8}$$

Since this initialization precedes weighted least squares, distances, residuals, and the cycle inconsistency $d_{ijk,t}$ are unavailable. Instead, $\tilde{s}_{ij}^*$ can be estimated using a variant of the update rule in (6), replacing residuals with previous $\tilde{s}_{ij,t}$ values and substituting $d_{ijk,t}$ with an AAB-style cycle inconsistency:

$$\tilde{d}_{ij,k} = \frac{1}{\pi}\min_{\boldsymbol{\gamma} \in S^2} d_g(\boldsymbol{\gamma}_{ij}, \boldsymbol{\gamma}) \in [0, 1],$$

where $\boldsymbol{\gamma}$ is constrained to satisfy

$$\alpha_{ij}\boldsymbol{\gamma} + \alpha_{jk}\boldsymbol{\gamma}_{jk}^* + \alpha_{ki}\boldsymbol{\gamma}_{ki}^* = \boldsymbol{0} \quad \text{for some } \alpha_{ij}, \alpha_{jk}, \alpha_{ki} > 0.$$

We refer the reader to [25] for a closed-form expression of $\tilde{d}_{ij,k}$. When computing $\tilde{s}_{ij}$, we use only well-shaped triangles-those where the angle between $\boldsymbol{\gamma}_{ik}$ and $\boldsymbol{\gamma}_{jk}$ lies in $[\arcsin(0.6), \pi-\arcsin(0.6)]$-since extreme angles can make $\tilde{d}_{ij,k}$ unstable, as discussed in our theory. We refer to this modified version of AAB as *truncated AAB (T-AAB)*.

Although this initialization cannot exactly estimate $\tilde{s}_{ij}^*$ for corrupted edges without location information, $\tilde{s}_{ij,t}$ is expected to approach zero on clean edges under the assumption of no additive noise. This allows for reliable separation of clean and corrupted edges, enabling exact location recovery.

Finally, we remark that Cycle-Sync is not sensitive to initialization. Nonetheless, the AAB-based initialization is lightweight, independent of least squares, and essentially a free improvement. A complete description of our location solver is given below.

In all of our experiments, we choose $t_{max} = 20$, $\beta = 20$ and $\lambda_t = t/(t+10)$. We remark that these parameters are not fine-tuned, but the performance of our method is already superior with the suboptimal parameters.

Our MPLS procedure differs from the original version for rotation estimation in several key aspects. First, unlike the rotation setting, we must address missing distance information, and our cycle-inconsistencies depend on iteratively updated distance estimates. Second, we adopt a Welsch-type objective, which better handles the large variation in residual scales typical in location estimation, whereas the original MPLS uses $\ell_{1/2}$ minimization. Lastly, our weighting parameter satisfies $\lambda_t \to 1$, gradually emphasizing cycle information over residuals, in contrast to the original one where $\lambda_t \to 0$. Further discussion and experiments on the annealing parameter $\lambda_t$ are included in Section I.3 in the supplementary material.

---

**Algorithm 1** Cycle-Sync

---

**Input:** $\{\boldsymbol{\gamma}_{ij}\}_{ij \in E}$, $\{\tilde{d}_{ij,k}\}_{k \in C_{ij}}$, $\beta$, $\{\lambda_t\}_{t \geq 1}$, $\delta$ (default: $10^{-8}$)
**Steps:**
Compute $\{s_{ij}\}_{ij \in E}$ by T-AAB
Initialize edge weights $w_{ij}^{(0)} = \exp(-20 s_{ij})$
**While** $t \leq t_{max}$**:**
    $t = t + 1$
    $\{\boldsymbol{t}_{i,t}\}_{i=1}^n, \{\alpha_{ij,t}\}_{ij \in E} = \arg\min_{\alpha_{ij} \geq 1, \sum_i \boldsymbol{t}_i = 0} \sum_{ij \in E} w_{ij,t} \|\boldsymbol{t}_i - \boldsymbol{t}_j - \alpha_{ij} \boldsymbol{\gamma}_{ij}\|^2$
    $r_{ij,t} = \|\boldsymbol{t}_{i,t} - \boldsymbol{t}_{j,t} - \alpha_{ij,t} \boldsymbol{\gamma}_{ij}\|$                         $ij \in E$
    $s_{ij,t} = \frac{1}{Z_{ij,t}} \sum_{k \in N_{ij}} e^{-\beta(r_{ik,t} + r_{jk,t})} \Big| \|\boldsymbol{t}_{i,t} - \boldsymbol{t}_{j,t}\| \boldsymbol{\gamma}_{ij} + \|\boldsymbol{t}_{j,t} - \boldsymbol{t}_{k,t}\| \boldsymbol{\gamma}_{jk} + \|\boldsymbol{t}_{k,t} - \boldsymbol{t}_{i,t}\| \boldsymbol{\gamma}_{ki} \Big|$
                                                                            $ij \in E$
    $h_{ij,t} = (1 - \lambda_t) r_{ij,t} + \lambda_t s_{ij,t}$                         $ij \in E$
    $w_{ij,t+1} = \frac{1}{4} f(h_{ij,t}) = \exp(-4 h_{ij,t})/(h_{ij,t} + \delta)$         $ij \in E$
**Output:** $\{\boldsymbol{t}_{i,t}\}_{i=1}^n$

---

## 2.3 Why we need cycle consistency: theory for exact location recovery

To motivate our use of cycle information, we present a theoretical result showing that AAB-style cycle inconsistency, when combined with iterative reweighting, can exactly separate clean and corrupted directions—even without access to inter-camera distances. Intuitively, this holds because corrupted directions tend to violate 3-cycle consistency, while clean directions remain geometrically consistent. Formally, under mild conditions, we show that the estimator $\tilde{s}_{ij,t}$ converges to zero on clean edges and remains bounded away from zero on corrupted ones, enabling *exact location recovery*. Under a probabilistic corruption model, our result also yields the *lowest known sample complexity* for exact recovery.

We assume that the edge set $E$ is partitioned into good (clean) edges $E_g$ and bad (corrupted) edges $E_b$. For $ij \in E_g$, $\boldsymbol{\gamma}_{ij} = \boldsymbol{\gamma}_{ij}^*$, and for $ij \in E_b$, $\boldsymbol{\gamma}_{ij}$ is an arbitrary unit vector distinct from $\boldsymbol{\gamma}_{ij}^*$. Define $N_{ij} = \{k \in [n] : ik, jk \in E\}$, $G_{ij} = \{k \in N_{ij} : ik, jk \in E_g\}$, and $B_{ij} = N_{ij} \setminus G_{ij}$. Let $\lambda = \max_{ij \in E} |B_{ij}|/|N_{ij}|$, $\mu = \min_{ij \in E_b} \sum_{k \in G_{ij}} \tilde{d}_{ij,k}/(|G_{ij}| s_{ij}^*)$ and $\theta_{ij,k}$ denote the angle between $\boldsymbol{\gamma}_{ik}$ and $\boldsymbol{\gamma}_{jk}$.

**Theorem 2.1.** *Assume there exists $\alpha > 0$ such that for all $ij \in E_g$ and $k \in N_{ij}$, $\alpha < \theta_{ij,k} < \pi - \alpha$, and $\lambda < 1 + eC_\alpha/\mu - \sqrt{eC_\alpha(2\mu + eC_\alpha)}/\mu$, where $C_\alpha = 2(\cos\alpha + \sqrt{5 - 4\cos^2\alpha})/\sin^2\alpha$. Then, for $\tilde{s}_{ij,t}$ computed by the iteratively reweighted AAB algorithm [25] using $\beta_0 \leq \frac{1}{2\lambda}$ and $\beta_{t+1} = r\beta_t$ with $1 < r < \mu(1-\lambda)^2/(2eC_\alpha\lambda)$, it holds for all $t > 0$ that*

$$\forall ij \in E_g, \quad \tilde{s}_{ij,t} \leq \frac{1}{2\beta_0 r^t}, \qquad \forall ij \in E_b, \quad \tilde{s}_{ij,t} \geq \frac{\mu}{e}(1-\lambda)\tilde{s}_{ij}^*.$$

In Theorem 2.1, the separation between clean and corrupted edges arises because the upper bound for clean edges, $1/(2\beta_0 r^t)$, vanishes as iteration $t \to \infty$ (note that $r > 1$), while the lower bound for bad edges remains strictly positive, proportional to their corruption levels. This result makes no assumptions on the distribution of corrupted directions, allowing fully adversarial and cycle-consistent

corruption. The angle condition on $\theta_{ij,k}$ addresses instability in $\tilde{d}_{ij,k}$ caused by ill-shaped triangles. To ensure this condition, we exclude triangles with $\theta_{ij,k} < \alpha$ or $\theta_{ij,k} > \pi - \alpha$ when computing $\tilde{d}_{ij,k}$. In the probabilistic setting with i.i.d. Gaussian locations, Erdős–Rényi edge probability $p$, and independent edge corruption probability $q$, our result achieves the strongest known recovery guarantee (see Table 1), where $n\epsilon_b$ is the maximum degree of the corrupted subgraph. The proof appears in the supplementary material.

| Method | $p$ | $\epsilon_b = pq$ |
|---|---|---|
| ShapeFit [7] | $\Omega(n^{-1/2}\log^{1/2} n)$ | $O(p^5/\log^3 n)$ |
| LUD [12] | $\Omega(n^{-1/3}\log^{1/3} n)$ | $O(p^{7/3}/\log^{9/2} n)$ |
| Our Theory (T-AAB) | $\Omega(n^{-1/2}\log^{1/2} n)$ | $O(p/\log^{1/2} n)$ |

Table 1: Phase transition bounds on $p$ (lower is better) and $\epsilon_b$ (higher is better) for location recovery.

## 2.4 Our full pipeline for camera pose estimation

We describe the full pipeline for camera pose estimation. Having covered location estimation, we focus here on the preceding steps: rotation and direction estimation. Given image keypoint matches (e.g., from SIFT or deep features), we estimate essential matrices for camera pairs using RANSAC, assuming calibrated cameras. The relative rotations $\boldsymbol{R}_{ij}$ are inferred from the essential matrices.

**Rotation averaging via MPLS-cycle.** Absolute rotations are recovered from relative ones using the MPLS algorithm. Unlike the original version, we fix $\lambda_t = 1$ to emphasize cycle consistency and disable IRLS reweighting. This variant, which we call *MPLS-cycle*, yields significantly lower orientation error on real SfM datasets. We report these results in the supplementary material.

**Direction estimation via robust subspace recovery.** Following [19], each ground-truth direction $\boldsymbol{\gamma}_{ij}^*$ is orthogonal to the vectors $\boldsymbol{v}_{ij}^k = \boldsymbol{R}_i \eta_i^k \times \boldsymbol{R}_j \eta_j^k$, where $\eta_i^k$ and $\eta_j^k$ are the normalized homogeneous coordinates of corresponding keypoints. Thus, $\boldsymbol{\gamma}_{ij}^*$ lies in the orthogonal complement of the subspace spanned by $\{\boldsymbol{v}_{ij}^k\}_k$. However, many $\boldsymbol{v}_{ij}^k$ vectors may be corrupted due to outlier matches. To robustly recover this subspace and estimate direction vectors (a problem reviewed in [9]), we use STE [35, 13], which significantly outperforms the REAPER method [10] used in the LUD pipeline [19].

**Outlier detection for directions.** As an optional filtering step, we use STE to reject direction estimates $\boldsymbol{\gamma}_{ij}$ with low inlier numbers. This plug-and-play module improves the accuracy of downstream location solvers. We observe notable performance gains with this preprocessing. In our real data experiment we use 20 as our minimum number of inliers.

## 3 Synthetic data experiments

We generate synthetic data under the uniform corruption model (UCM), where $n$ is the number of cameras, $q$ is the corruption probability per edge, $\sigma$ is the noise level, and $p$ is the probability of edge connection in the Erdős–Rényi viewing graph. Ground-truth camera locations $\boldsymbol{t}_i^*$ are sampled independently from $N(\boldsymbol{0}, \boldsymbol{I}_{3\times3})$. For each edge $ij$ in the generated graph, the observed direction is given by:

$$\boldsymbol{\gamma}_{ij} = \begin{cases} \text{Normalize}(\boldsymbol{t}_i^* - \boldsymbol{t}_j^* + \sigma\boldsymbol{\epsilon}_{ij}), & \text{w.p. } 1-q; \\ \text{Normalize}(\boldsymbol{\epsilon}_{ij}), & \text{w.p. } q, \end{cases} \tag{9}$$

where $\boldsymbol{\epsilon}_{ij} \sim N(\boldsymbol{0}, \boldsymbol{I}_{3\times3})$ independently.

We also consider an adversarial corruption model where corrupted directions remain cycle-consistent:

$$\boldsymbol{\gamma}_{ij} = \begin{cases} \text{Normalize}(\boldsymbol{t}_i^* - \boldsymbol{t}_j^* + \sigma\boldsymbol{\epsilon}_{ij}), & \text{w.p. } 1-q; \\ \text{Normalize}(\boldsymbol{t}_i^c - \boldsymbol{t}_j^c + \sigma\boldsymbol{\epsilon}_{ij}), & \text{w.p. } q, \end{cases} \tag{10}$$

where $\{\boldsymbol{t}_i^c\}$ is a set of alternative locations used to generate coherent corrupted directions. To avoid ambiguity with the true structure, we require $q < 0.5$ in this model.

We also test the robustness to corruption for different methods. We fix $n = 100$, $p = 0.5$. We obtain the estimated absolute locations $\hat{\boldsymbol{t}}_i$ by our Cycle-Sync solver and compare the estimation error with

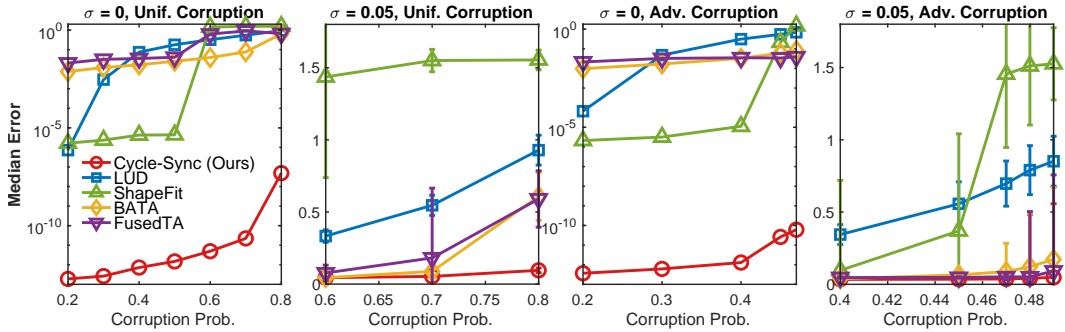

Figure 3: Median camera location error versus corruption probability. Left to right: (1) uniform corruption without noise, (2) uniform corruption with mild noise ($\sigma = 0.05$), (3) cycle-consistent (adversarial) corruption without noise, and (4) cycle-consistent corruption with mild noise. Error bars indicate standard deviation over 10 independent trials. No error bars are shown in plots with logarithmic scale.

existing works ShapeFit [5], BATA [36], LUD [19], FusedTA [15]. Since absolute locations can only be estimated up to a global translation and scaling, we remove these ambiguities by computing the minimizer $(c^*, t^*)$ of $\min_{c \in \mathbb{R}, t \in \mathbb{R}^3} \sum_{i \in [n]} \|t_i^* - (c\hat{t}_i + t)\|_2$.

We compute the absolute translation error for camera $i$ as $\|t_i^* - (c^*\hat{t}_i + t^*)\|_2$ and report the median error over all cameras in Figure 3. We generate 10 instances of the synthetic data and report the standard deviation of these statistics with error bars (no error bars with log scales).

We say a method achieves "exact recovery" if it estimates the absolute camera locations with less than $10^{-4}$ median error. On UCM with $\sigma = 0$, exact recovery is achieved only when $q \leq 0.3$ except our method and ShapeFit, while our method exactly recovers when $q \leq 0.8$. Therefore, our method improves the phase transition threshold for exact recovery by a large margin. For the adversarial setting, Cycle-Sync remains robust up to corruption rates near the theoretical limit ($q < 0.5$), while all baselines quickly deteriorate. These results validate the effectiveness of our cycle-based reweighting and its ability to leverage global consistency for accurate location recovery in both stochastic and adversarial scenarios.

## 4    Real data experiments

We conduct real-world experiments on 13 ETH3D stereo datasets [23, 24], using a personal laptop equipped with an 11th Gen Intel(R) Core(TM) i9-11900H processor (2.50 GHz, 8 cores, 16 threads) and 16 GB physical memory. This dataset contains 13 sets of undistorted images taken from different indoor and outdoor scenes. It is challenging because of its occlusions, viewpoint changes and lighting differences which results in highly corrupted pairwise directions. From raw images, we estimate the initial keypoint matches using SIFT feature matching and perform geometric verification with RANSAC. With these initial keypoint matches as the input, we go through our full camera pose estimation pipeline and compute the output absolute location estimates $\{\hat{t}_i\}_{i \in [n]}$ and the absolute rotation estimates $\{\hat{R}_i\}_{i \in [n]}$. We remark that if the viewing graph contains multiple weakly connected components that cannot be merged into a single scene, we retain only the largest component for rotation and location estimation, and exclude the others from computation and evaluation.

We use the millimeter-accurate, laser-scanned camera poses from the dataset as the ground truth. To eliminate the translation and scale ambiguity in camera locations, we preprocess the camera locations by translating them so that the geometric mean of all camera positions is centered at the origin, followed by scaling to ensure that the median distance of the cameras from the origin is 1. We name the preprocessed ground truth camera locations as $\{t_i^*\}_{i \in [n]}$ and camera orientations as $\{R_i^*\}_{i \in [n]}$. To evaluate the output, we first align the rotation estimates with $R_{\text{align}}$, the minimizer of the L2 rotation alignment error $\min_{R \in \mathbb{R}^{3 \times 3}} \sum_{i \in [n]} \|R_i^* - RR_i\|_F^2$. Then, to remove the global scale and translation ambiguities for camera locations, we compute the minimizer of the L1 alignment

error as $(c^*, \boldsymbol{t}^*)$: $\min_{c \in \mathbb{R}, t \in \mathbb{R}^3} \sum_{i \in [n]} \| \boldsymbol{t}_i^* - (c \boldsymbol{R}_{\text{align}} \hat{\boldsymbol{t}}_i + \boldsymbol{t}) \|_2$. For each camera $i$, we compute the translation error as $\| \boldsymbol{t}_i^* - (c^* \boldsymbol{R}_{\text{align}} \hat{\boldsymbol{t}}_i + \boldsymbol{t}^*) \|_2$. For each dataset, we report the median error over cameras. We test our method Cycle-Sync, BATA [36], FusedTA [15], ShapeFit [5] and LUD [19] for comparison of different location estimation methods, while the camera orientation method is fixed to MPLS-Cycle. Also, to compare with existing global SfM pipelines, we report the translation error of camera poses from LUD, Theia [30] and GLOMAP [22]. In addition, we conduct ablation studies. Starting from LUD+IRLS, the original LUD pipeline, we gradually add each building block of our full pipeline to demonstrate the effectiveness of each block. We summarize the results in Figure 4. We highlight some key comparisons derived from our experiments.

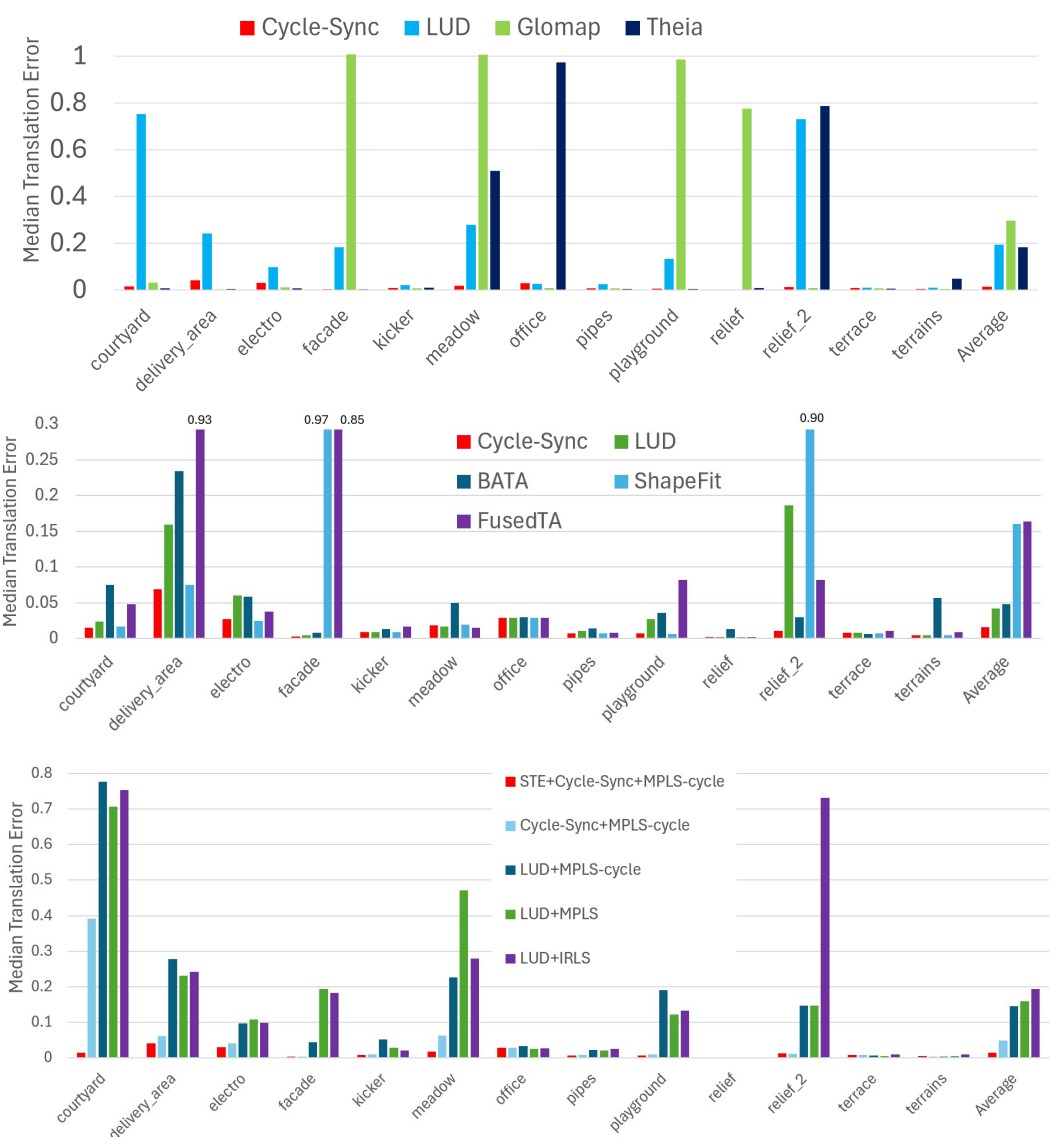

Figure 4: Median translation errors using ETH3D. Each column represents a dataset, the last one shows the average median error across all datasets. Different methods are presented per column. **Upper:** comparison of all pipelines. Unlike Theia and GLOMAP, Cycle-Sync (ours) estimates locations without bundle adjustment. **Middle:** comparison for different camera location algorithms; all methods preprocessed by STE and MPLS-cycle for fair comparison. **Lower:** Ablation studies.

**Comparison among SfM pipelines (top panel).** We observe that our full pipeline significantly outperforms existing SfM pipelines when averaging across datasets. Here, LUD refers to the original unmodified version proposed in the literature. Our method reduces the median location error to below 0.05, whereas other pipelines yield considerably less accurate results, with median errors exceeding 0.2 on average. Despite bypassing bundle adjustment, our method yields more accurate camera locations on average, whereas Theia and GLOMAP, even with bundle adjustment, suffer from many outliers and large alignment errors on several datasets. Although Theia performs better on the majority of datasets, our method consistently avoids failure cases and achieves the lowest average error across all datasets. Both metrics—the number of datasets with superior performance and the average performance across all datasets—are informative: the former reflects robustness across the majority of scenarios, while the latter highlights stability and resilience against more challenging or adversarial datasets.

**Comparison of location estimation algorithms (middle panel).** In this comparison, we fix all preceding steps across all location solvers for fairness. Specifically, for all baselines, we employ MPLS-Cycle for rotation estimation and STE for direction estimation and filtering. Under this standardized setup, our method consistently outperforms others on 10 out of the 14 datasets. In terms of median location error averaged over all datasets, our approach (with STE) achieves a reduction of 60.9% compared to LUD (with STE), 66.1% compared to BATA, 89.8% compared to ShapeFit, and 90.0% compared to FusedTA.

**Ablation studies (bottom panel).** We observe that each component of our method consistently reduces the median location error. Starting from the LUD pipeline (LUD for location + REAPER for pairwise direction + IRLS [3] for rotation averaging), upgrading IRLS to MPLS reduces median location error by 17.8%; upgrading MPLS to MPLS-cycle reduces location error further by 9.0%. This is because better camera orientations improve the initial pairwise direction estimates. We remark that STE is an optional module for Cycle-Sync. Cycle-Sync outperforms all baselines (under the same preprocessing) both with and without STE. Specifically, upgrading LUD (without STE) to Cycle-Sync (without STE) reduces the location error by 66.0%, and the above panel indicates error reduction by 60.9% when using STE for both LUD and Cycle-Sync. Finally, replacing REAPER with STE within Cycle-Sync yields a 70.5% improvement. This significant gain stems from STE's effectiveness not just as an algorithm, but also from our specific use of STE in filtering outlying directions.

For other details, we refer the readers to the table in the supplementary material.

## 5    Conclusion and Limitations

We proposed a global framework for camera pose estimation that fully exploits cycle-consistency information. Our method simultaneously addresses the challenges posed by large variations in pairwise distances and highly corrupted directions. We establish the strongest known exact recovery guarantee for location estimation under adversarial corruption, which also yields improved sample complexity under standard probabilistic models. Empirically, our method is the only algorithm that consistently outperforms all state-of-the-art baselines on real datasets.

Several limitations remain. Our theoretical guarantees apply only to the initialization phase and do not extend to the convergence of the full nonconvex optimization procedure. Establishing guarantees for the entire location synchronization algorithm—and eventually the full pose estimation pipeline—is an important direction for future work. In addition, the method relies on the presence of well-shaped 3-cycles, and performance may degrade on sparse or structured graphs lacking such cycles. All hyperparameters (e.g., reweighting schedules and robust loss parameters) are manually selected, and while not overly sensitive, it would be valuable to develop more adaptive or learnable reweighting strategies.

**Broader Impact** Our work advances the robustness and theoretical understanding of structure-from-motion (SfM) pipelines, with potential applications in robotics, autonomous navigation, digital reconstruction of cultural heritage, and low-cost 3D mapping. By improving pose estimation under high corruption and without bundle adjustment, our method may make SfM more accessible and reliable in challenging or resource-constrained environments.

**Acknowledgement**    G. Lerman and S. Li were supported by NSF DMS-2152766. Y. Shi was supported by NSF DMS-2514152.

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

# Supplementary Material

We provide additional theoretical and experimental details that complement the main paper. In Section A, we present full proofs of Theorem 2.1. Section B explains the sample complexity analysis that underlies the bounds for T-AAB reported in Table 1. Sections C–G provide additional experimental results: 3D reconstructions (Section C and D), supplementary figures to the ETH3D dataset (Section E), runtime comparisons (Section F), and extended results on rotation synchronization (Section G), and generalization to the IMC-PT dataset (Section H).

All equation, figure, table, and theorem numbers continue from the main paper.

## A  Proofs of theory

We first establish Theorem A.1 and then conclude the main theorem.

**Theorem A.1.** *Assume there exists an absolute $\alpha > 0$ such that for any $ij \in E_g$ and $k \in N_{ij}$, $\alpha < \theta_{ij,k} < \pi - \alpha$. Then for $ij \in E_g$, we have $\tilde{d}_{ij,k} \leq C_\alpha(\tilde{s}_{ik}^* + \tilde{s}_{jk}^*)$, where $C_\alpha = \frac{2(\cos\alpha + \sqrt{5 - 4\cos^2\alpha})}{\sin^2\alpha}$.*

We note that the triangle is ill-shaped whenever $\theta_{ij,k} \approx 0$ or $\theta_{ij,k} \approx \pi$. In practice, we want $d_{ij,k}$ to be small for a clean edge $ij \in E_g$ whenever the other two edges are relatively clean with small $\tilde{s}_{ik}^*$ and $\tilde{s}_{jk}^*$. However, in these two ill-shaped cases, $C_\alpha$ in the theorem goes to infinity, and there is no effective upper bound to control $\tilde{d}_{ij,k}$.

**Proof.** Let $\gamma_p$ be the projected vector of $\gamma_{ij}$ onto $\mathrm{Span}(\gamma_{ik}, \gamma_{kj})$. Denote $x = \gamma_{ij}^T \gamma_{ki}$, $y = \gamma_{ij}^T \gamma_{jk}$ and $z = \gamma_{jk}^T \gamma_{ki}$. Since $\gamma_p$ is in $\mathrm{Span}(\gamma_{ki}, \gamma_{kj})$, there exists constants $a, b$ such that $\gamma_p = a\gamma_{ki} + b\gamma_{jk}$. By the definition of $\gamma_p$, we have

$$\begin{cases} \langle \gamma_{ij} - a\gamma_{ki} - b\gamma_{jk}, \gamma_{ki} \rangle = 0 \\ \langle \gamma_{ij} - a\gamma_{ki} - b\gamma_{jk}, \gamma_{jk} \rangle = 0 \end{cases} . \tag{11}$$

By linearity of vector inner products, we have

$$\begin{cases} x - a - bz = 0 \\ y - az - b = 0 \end{cases} . \tag{12}$$

Solving for $a$ and $b$ gives

$$\begin{cases} a = \frac{x - yz}{1 - z^2} \\ b = \frac{y - xz}{1 - z^2} \end{cases} . \tag{13}$$

Therefore $\gamma_p = \frac{x - yz}{1 - z^2}\gamma_{ki} + \frac{y - xz}{1 - z^2}\gamma_{jk}$.

**Case 1:** $\gamma_p \notin \Omega(\gamma_{jk}, \gamma_{ki})$. In this case, since $\tilde{d}_{ij,k} \leq 1$, we only need to prove $\tilde{s}_{ik}^* + \tilde{s}_{jk}^* > 1/C_\alpha$.

Note that by the definition of $\Omega(\gamma_{jk}, \gamma_{ki})$ we have either $a < 0$ or $b < 0$, which implies $x - yz > 0$ or $y - xz > 0$. On the other hand, since the ground truth directions $\gamma_{ij}^*, \gamma_{jk}^*, \gamma_{ki}^*$ are cycle consistent, we know that the projection of $\gamma_{ij}^*$ onto $\mathrm{Span}(\gamma_{ki}^*, \gamma_{jk}^*)$ is in the set $\Omega(\gamma_{ki}^*, \gamma_{jk}^*)$. Therefore we also have $x^* - y^* z^* < 0$ and $y^* - x^* z^* < 0$, where $x^* = \gamma_{ij}^T \gamma_{ki}^*$, $y^* = \gamma_{ij}^T \gamma_{jk}^*$ and $z^* = \gamma_{jk}^{*T} \gamma_{ki}^*$. Without loss of generality, we assume the case $x - yz > 0$. If $\max(\tilde{s}_{ik}^*, \tilde{s}_{jk}^*) \geq \frac{1}{C_\alpha}$, then the lemma is trivial. If $\max(\tilde{s}_{ik}^*, \tilde{s}_{jk}^*) < \frac{1}{C_\alpha}$, we first verify two claims.

**Claim 1:** $x^* - y^* z^* < -\sin\alpha^*$, where $\alpha^*$ is the angle such that $\cos\alpha^* = \cos\alpha + \frac{2}{C_\alpha}$.

By the definition of $\tilde{s}_{ik}^*$ and $\tilde{s}_{jk}^*$, we have the following inequality:

$$|\gamma_{ik} - \gamma_{ik}^*| + |\gamma_{jk} - \gamma_{jk}^*| = 2\sin\frac{\tilde{s}_{ik}^*}{2} + 2\sin\frac{\tilde{s}_{jk}^*}{2} \leq \tilde{s}_{ik}^* + \tilde{s}_{jk}^* \leq \frac{2}{C_\alpha}. \tag{14}$$

Also, $\alpha < \theta_{ij,k} < \pi - \alpha$ is equivalent to $|\gamma_{jk}^T \gamma_{ki}| = |\cos\theta_{ij,k}| < \cos\alpha$. Combining this with equation (14) gives

$$
\begin{aligned}
|\gamma_{jk}^{*T}\gamma_{ki}^*| &= |\gamma_{jk}^T\gamma_{ki} + (\gamma_{jk}^* - \gamma_{jk})^T\gamma_{ki} + \gamma_{jk}^{*T}(\gamma_{ki}^* - \gamma_{ki})| \\
&\le |\gamma_{jk}^T\gamma_{ki}| + |(\gamma_{jk}^* - \gamma_{jk})^T\gamma_{ki}| + |\gamma_{jk}^{*T}(\gamma_{ki}^* - \gamma_{ki})| \\
&\le |\gamma_{jk}^T\gamma_{ki}| + |\gamma_{jk}^* - \gamma_{jk}| + |\gamma_{ki}^* - \gamma_{ki}| \\
&\le \cos\alpha + \frac{2}{C_\alpha} = \cos\alpha^*.
\end{aligned}
\tag{15}
$$

On the other hand, we know that $a^* = \frac{x^* - y^*z^*}{1 - z^{*2}}$, and $a^* = -|a^*| \le -\sin\alpha^*$. This implies that $x^* - y^*z^* < -\frac{\sin\alpha^*}{1} = -\sin\alpha^*$.

**Claim 2:** Let $\delta_{ij} = \gamma_{ij} - \gamma_{ij}^*$, $\delta_{jk} = \gamma_{jk} - \gamma_{jk}^*$ and $\delta_{ki} = \gamma_{ki} - \gamma_{ki}^*$. Suppose $\max(|\delta_{ij}|, |\delta_{jk}|, |\delta_{ki}|) = \delta$. Then $|(x^* - y^*z^*) - (x - yz)| \le 6\delta$; if $ij \in E_g$ (i.e. $\delta_{ij} = 0$), then $|(x^* - y^*z^*) - (x - yz)| \le 4\delta$.

In fact, by the definition of $x, x^*, y, y^*, z, z^*$, we have the following estimate:

$$
\begin{aligned}
|(x^* - y^*z^*) - (x - yz)| &= |(\gamma_{ij}^{*T}\gamma_{ki}^* - \gamma_{ij}^{*T}\gamma_{jk}^*\gamma_{jk}^{*T}\gamma_{ki}^*) - ((\gamma_{ij}^* + \delta_{ij})^T(\gamma_{ki}^* + \delta_{ki}) \\
&\quad - (\gamma_{ij}^* + \delta_{ij})^T(\gamma_{jk}^* + \delta_{jk})(\gamma_{jk}^* + \delta_{jk})^T(\gamma_{ki}^* + \delta_{ki}))| \\
&\le |-\delta_{ij}^T\gamma_{ki} - \gamma_{ij}^{*T}\delta_{ki} - (\delta_{ij}^T\gamma_{jk}\gamma_{jk}^T\gamma_{ki} + \gamma_{ij}^{*T}\delta_{jk}\gamma_{jk}^T\gamma_{ki} \\
&\quad + \gamma_{ij}^{*T}\gamma_{jk}^*\delta_{jk}^T\gamma_{ki} + \gamma_{ij}^{*T}\gamma_{jk}^*\gamma_{jk}^T\delta_{ki})| \\
&\le |\delta_{ij}^T\gamma_{ki}| + |\gamma_{ij}^{*T}\delta_{ki}| + |\delta_{ij}^T\gamma_{jk}\gamma_{jk}^T\gamma_{ki}| + |\gamma_{ij}^{*T}\delta_{jk}\gamma_{jk}^T\gamma_{ki}| \\
&\quad + |\gamma_{ij}^{*T}\gamma_{jk}^*\delta_{jk}^T\gamma_{ki}| + |\gamma_{ij}^{*T}\gamma_{jk}^*\gamma_{jk}^T\delta_{ki}|.
\end{aligned}
\tag{16}
$$

By the fact that all $\gamma$'s are unit vectors, the right hand side of the equation above is at most $6\delta$ in general; if $ij \in E_g$ (i.e. $\delta_{ij} = 0$) then it is at most $4\delta$.

Combining claim 1 and claim 2, we know that $0 < x - yz \le (x^* - y^*z^*) + |(x - yz) - (x^* - y^*z^*)| \le 4\delta - \sin\alpha^*$. This yields $\delta > \frac{\sin\alpha^*}{4}$. Note that by $ij \in E_g$, we know that $\delta_{ij} = 0$. Therefore $\delta = \max(|\delta_{ij}|, |\delta_{jk}|, |\delta_{ki}|) = \max(|\delta_{jk}|, |\delta_{ki}|) \le |\delta_{jk}| + |\delta_{ki}| = 2\sin\frac{\tilde{s}_{jk}^*}{2} + 2\sin\frac{s_{ki}^*}{2} \le \tilde{s}_{jk}^* + s_{ki}^*$. By $C_\alpha = \frac{2(\cos\alpha + \sqrt{5 - 4\cos^2\alpha})}{\sin^2\alpha}$, we know that $\frac{\sin\alpha^*}{4} = \frac{1}{C_\alpha}$, therefore the theorem is proved.

**Case 2:** $\gamma_p \in \Omega(\gamma_{ik}, \gamma_{kj})$. In this case, let $\delta_{ik} = \gamma_{ik} - \gamma_{ik}^*$ and $\delta_{jk} = \gamma_{jk} - \gamma_{jk}^*$. Then $\tilde{d}_{ij,k} = \frac{|\gamma_{ik} \times \gamma_{kj} \cdot \gamma_{ij}|}{\sin\theta_{ikj}}$. By the fact that $\gamma_{ij}^*, \gamma_{jk}^*, \gamma_{ki}^*$ are coplanar and $\gamma_{ij} = \gamma_{ij}^*$, we know that $\gamma_{ik}^* \times \gamma_{kj}^* \cdot \gamma_{ij} = 0$. We have the following inequalities:

$$
\begin{aligned}
\tilde{d}_{ij,k} &= \frac{|\gamma_{ik} \times \gamma_{kj} \cdot \gamma_{ij}|}{\sin\theta_{ij,k}} \\
&= \frac{|(\gamma_{ik}^* + \delta_{ik}) \times (\gamma_{kj}^* + \delta_{kj}) \cdot \gamma_{ij}|}{\sin\theta_{ij,k}} \\
&= \frac{|(\delta_{ik} \times (\gamma_{kj}^* + \delta_{kj}) + \gamma_{ik}^* \times \delta_{kj}) \cdot \gamma_{ij}|}{\sin\theta_{ij,k}} \\
&\le \frac{|\delta_{ik}||\gamma_{kj}^* + \delta_{kj}| + |\gamma_{ik}^*||\delta_{kj}|}{\sin\theta_{ij,k}} \tag{17} \\
&\le \frac{\delta_{ik} + \delta_{kj}}{\sin\alpha}. \tag{18}
\end{aligned}
$$

Note that $|\delta_{ik}| = 2\sin\frac{\tilde{s}_{ik}^*}{2} \le \tilde{s}_{ik}^*$, and similarly $|\delta_{jk}| = 2\sin\frac{\tilde{s}_{jk}^*}{2} \le \tilde{s}_{jk}^*$. Therefore $\tilde{d}_{ij,k} \le \frac{1}{\sin\alpha}(\tilde{s}_{ik}^* + \tilde{s}_{jk}^*) \le C_\alpha(\tilde{s}_{ik}^* + \tilde{s}_{jk}^*)$, where the latter inequality comes from the fact that $C_\alpha \ge \frac{1}{\sin\alpha}$.

**Proof of the main theorem.**

Recall the statement of Theorem 2.1:

**Main Theorem** (Theorem 2.1). *Assume there exists $\alpha > 0$ such that for all $ij \in E_g$ and $k \in N_{ij}$, $\alpha < \theta_{ij,k} < \pi - \alpha$, and $\lambda < 1 + eC_\alpha/\mu - \sqrt{eC_\alpha(2\mu + eC_\alpha)}/\mu$, where $C_\alpha = 2(\cos\alpha + \sqrt{5 - 4\cos^2\alpha})/\sin^2\alpha$. Then, for $\tilde{s}_{ij,t}$ computed by the iteratively reweighted AAB algorithm [25] using $\beta_0 \leq \frac{1}{2\lambda}$ and $\beta_{t+1} = r\beta_t$ with $1 < r < \mu(1-\lambda)^2/(2eC_\alpha\lambda)$, it holds for all $t > 0$ that*

$$\forall ij \in E_g, \quad \tilde{s}_{ij,t} \leq \frac{1}{2\beta_0 r^t}, \qquad \forall ij \in E_b, \quad \tilde{s}_{ij,t} \geq \frac{\mu}{e}(1-\lambda)\tilde{s}_{ij}^*.$$

**Proof**. We prove the main theorem by induction. For $t = 0$, the definition of $\lambda$ imply that for all $ij \in E$,

$$\tilde{s}_{ij}^{(0)} = \frac{\sum_{k \in N_{ij}} \tilde{d}_{ij,k}}{|N_{ij}|} \geq \frac{\sum_{k \in G_{ij}} \tilde{d}_{ij,k}}{|N_{ij}|} \geq \mu\frac{|G_{ij}|}{|N_{ij}|}\tilde{s}_{ij}^* \geq \mu(1-\lambda)\tilde{s}_{ij}^*.$$

Furthermore, by the fact that $0 \leq \tilde{d}_{ij,k} \leq 1$ we have for all $ij \in E_g$,

$$\tilde{s}_{ij}^{(0)} = \frac{\sum_{k \in N_{ij}} \tilde{d}_{ij,k}}{|N_{ij}|} = \frac{\sum_{k \in B_{ij}} \tilde{d}_{ij,k}}{|N_{ij}|} \leq \frac{\sum_{k \in B_{ij}} 1}{|N_{ij}|} \leq \lambda \leq \frac{1}{2\beta_0}.$$

Therefore the theorem is proved when $t = 0$.

Next, we assume the theorem holds true for $0, 1, \cdots, t$, and show that it also holds true for $t + 1$. By the definition of $\tilde{s}_{ij}^{(t+1)}$ and the induction assumption $\frac{1}{2\beta_t} \geq \max_{ij \in E_g} \tilde{s}_{ij,t}$, we have the following inequalities for any $ij \in E_b$:

$$
\begin{aligned}
\tilde{s}_{ij}^{(t+1)} &= \frac{\sum_{k \in N_{ij}} e^{-\beta_t(\tilde{s}_{ik}^{(t)} + \tilde{s}_{jk}^{(t)})}\tilde{d}_{ij,k}}{\sum_{k \in N_{ij}} e^{-\beta_t(\tilde{s}_{ik}^{(t)} + \tilde{s}_{jk}^{(t)})}} \\
&\geq \frac{\sum_{k \in G_{ij}} e^{-\beta_t(\tilde{s}_{ik}^{(t)} + \tilde{s}_{jk}^{(t)})}\tilde{d}_{ij,k}}{\sum_{k \in N_{ij}} e^{-\beta_t(\tilde{s}_{ik}^{(t)} + \tilde{s}_{jk}^{(t)})}} \\
&\geq \frac{\sum_{k \in G_{ij}} e^{-1}\tilde{d}_{ij,k}}{|N_{ij}|} \\
&\geq \frac{\mu}{e}\frac{|G_{ij}|}{|N_{ij}|}\tilde{s}_{ij}^* \\
&\geq \frac{\mu(1-\lambda)}{e}\tilde{s}_{ij}^*.
\end{aligned}
\tag{19}
$$

Next we bound $\tilde{s}_{ij}^{(t+1)}$ for $ij \in E_g$. By the definition of $\tilde{s}_{ij}^{(t+1)}$, the fact that $\tilde{d}_{ij,k} = 0$ for $k \in G_{ij}$, and Theorem A.1 we know that

$$
\tilde{s}_{ij}^{(t+1)} = \frac{\sum_{k \in N_{ij}} e^{-\beta_t(\tilde{s}_{ik}^{(t)} + \tilde{s}_{jk}^{(t)})}\tilde{d}_{ij,k}}{\sum_{k \in N_{ij}} e^{-\beta_t(\tilde{s}_{ik}^{(t)} + \tilde{s}_{jk}^{(t)})}} = \frac{\sum_{k \in B_{ij}} e^{-\beta_t(\tilde{s}_{ik}^{(t)} + \tilde{s}_{jk}^{(t)})}\tilde{d}_{ij,k}}{\sum_{k \in N_{ij}} e^{-\beta_t(\tilde{s}_{ik}^{(t)} + \tilde{s}_{jk}^{(t)})}}
\tag{20}
$$

$$
\leq \frac{C_\alpha \sum_{k \in B_{ij}} e^{-\beta_t(\tilde{s}_{ik}^{(t)} + \tilde{s}_{jk}^{(t)})}(\tilde{s}_{ik}^* + \tilde{s}_{jk}^*)}{\sum_{k \in N_{ij}} e^{-\beta_t(\tilde{s}_{ik}^{(t)} + \tilde{s}_{jk}^{(t)})}}.
\tag{21}
$$

By the induction assumption that $\tilde{s}_{ij}^{(t)} \geq \frac{\mu(1-\lambda)}{e}\tilde{s}_{ij}^*$ for all $ij \in E$, we know that

$$
\sum_{k \in B_{ij}} e^{-\beta_t(\tilde{s}_{ik}^{(t)} + \tilde{s}_{jk}^{(t)})}(\tilde{s}_{ik}^* + \tilde{s}_{jk}^*) \leq \sum_{k \in B_{ij}} e^{-\beta_t\frac{\mu(1-\lambda)}{e}(\tilde{s}_{ik}^* + \tilde{s}_{jk}^*)}(\tilde{s}_{ik}^* + \tilde{s}_{jk}^*).
\tag{22}
$$

Note that $xe^{-cx} < \frac{1}{ce}$ for any $c > 0$ and $x > 0$. Let $c = \beta_t \frac{\mu(1-\lambda)}{e}$ and $x = \tilde{s}_{ik}^* + \tilde{s}_{jk}^*$, we have

$$\sum_{k \in B_{ij}} e^{-\beta_t \frac{\mu(1-\lambda)}{e}(\tilde{s}_{ik}^* + \tilde{s}_{jk}^*)}(\tilde{s}_{ik}^* + \tilde{s}_{jk}^*) \leq \sum_{k \in B_{ij}} \frac{1}{\beta_t \mu(1-\lambda)} = \frac{|B_{ij}|}{\beta_t \mu(1-\lambda)}. \tag{23}$$

Also, by the induction assumption that $\frac{1}{2\beta_t} \geq \max_{ij \in E_g} \tilde{s}_{ij}^{(t)}$ and the nonnegativity of $\tilde{s}_{ij}^{(t)}$'s, we have

$$\sum_{k \in N_{ij}} e^{-\beta_t(\tilde{s}_{ik}^{(t)} + \tilde{s}_{jk}^{(t)})} \geq \sum_{k \in G_{ij}} e^{-\beta_t(\tilde{s}_{ik}^{(t)} + \tilde{s}_{jk}^{(t)})} \geq |G_{ij}|e^{-1}. \tag{24}$$

Combining (20), (22), (23), (24) and the definition of $\lambda$, we have

$$\tilde{s}_{ij}^{(t+1)} \leq \frac{|B_{ij}|}{|G_{ij}|} \cdot \frac{eC_\alpha}{\beta_t \mu(1-\lambda)} \leq \frac{2eC_\alpha \lambda}{\mu(1-\lambda)^2} \cdot \frac{1}{2\beta_t}. \tag{25}$$

By the assumption that $\lambda < 1 + \frac{eC_\alpha}{\mu} - \sqrt{\frac{eC_\alpha}{\mu}(2 + \frac{eC_\alpha}{\mu})}$, we know that $\frac{2e\lambda}{\mu(1-\lambda)^2} < 1$. Therefore by taking $1 < r < \frac{\mu(1-\lambda)^2}{2e\lambda}$, we guarantee that for any $ij \in E_g$, $\tilde{s}_{ij}^{(t+1)} \leq \frac{1}{2\beta_{t+1}} = \frac{1}{2\beta_0 r^{t+1}}$. This and (19) concludes our theorem. $\square$

**Comment on the order of $\mu$:** We remark that in the theorem $\mu = \min_{ij \in E_b} \sum_{k \in G_{ij}} \tilde{d}_{ij,k}/(|G_{ij}|\tilde{s}_{ij}^*)$, which implies that for all $ij \in E$,

$$\frac{1}{|G_{ij}|} \sum_{k \in G_{ij}} \tilde{d}_{ij,k} \geq \mu \tilde{s}_{ij}^*. \tag{26}$$

We would like to investigate the dependence of $\mu$ on $n$. That is, we estimate the magnitude of $\mu$ such that (26) holds for all edges. First of all, it is safe to claim that (26) holds for all $ij$ whose $\tilde{s}_{ij}^* > 0.5$ when $\mu$ is a positive constant (i.e., $\mu = \Theta(1)$). That is, the left-hand side of (26) is lower bounded by a positive constant. Let $\boldsymbol{n}_{ij,k}$ be the normal vector of the plane $\mathrm{Span}\{\boldsymbol{t}_k^* - \boldsymbol{t}_i^*, \boldsymbol{t}_k^* - \boldsymbol{t}_j^*\}$, where $\boldsymbol{t}_i^*$ follows the standard Gaussian distribution. For $\tilde{s}_{ij}^* \leq 0.5$, one can show that

$$\frac{1}{|G_{ij}|} \sum_{k \in G_{ij}} \tilde{d}_{ij,k} \geq \frac{1}{|G_{ij}|} \sum_{k \in G_{ij}} |\boldsymbol{\gamma}_{ij}^\top \boldsymbol{n}_{ij,k}| \geq \frac{c'}{\sqrt{\log n}}|\boldsymbol{\gamma}_{ij}^\top \boldsymbol{\gamma}_{ij}^*|$$
$$= \frac{c}{\sqrt{\log n}} \min\left(\tilde{s}_{ij}^*, 1 - \tilde{s}_{ij}^*\right) = \frac{c}{\sqrt{\log n}} \tilde{s}_{ij}^*, \tag{27}$$

for some absolute constant $c', c$, which suggests

$$\mu \geq c/\sqrt{\log n}.$$

In (27), the first inequality follows from the definition of $\tilde{d}_{ij,k}$, the first equality follows from the definition of $\tilde{s}_{ij}^*$ and the last equality is due to the assumption $\tilde{s}_{ij}^* \leq 0.5$. The second inequality is commonly assumed for all $ij \in E$ in [7, 12, 25], which they call the $c/\sqrt{\log n}$-well-distributed condition. It is proved in [7] that the if $\boldsymbol{t}_i^*$ is i.i.d. with standard Gaussian, then $c/\sqrt{\log n}$-well-distributed condition holds with high probability.

## B  Explanation of the Order of Complexity for T-AAB in Table 1

We assume the Erdős–Rényi graph $G(n, p)$, where $p$ is the probability of connecting two nodes, with edge corruption probability $q$. We show that the recovery guarantee in Theorem 2.1 holds under this probabilistic model, provided

$$p = \Omega(n^{-1/2} \log^{1/2} n) \tag{28}$$

and

$$\epsilon_b = pq = O(p/\sqrt{\log n}). \tag{29}$$

We note that given (28), (29) is equivalent to

$$q = O(1/\sqrt{\log n}). \tag{30}$$

We first verify (30), where we note that it is sufficient to focus on the worst case

$$q = c_1/\sqrt{\log n} \quad \text{for an absolute constant } c_1.$$

That is, we show that this choice is sufficient for exact recovery with high probability.

We prove exact recovery by establishing with high probability the sufficient condition of Theorem 2.1:

$$\lambda < 1 + \frac{eC_\alpha}{\mu} - \sqrt{\frac{eC_\alpha}{\mu}\left(2 + \frac{eC_\alpha}{\mu}\right)}. \tag{31}$$

We control the ratio of bad cycles as follows. For any fixed edge $ij \in E$, $\lambda_{ij} = |B_{ij}|/|N_{ij}|$ is the average of the Bernoulli random variables $X_k = \mathbf{1}_{\{k \in B_{ij}\}}$ where $k \in B_{ij}$ with probability $1 - (1-q)^2$. Consequently,

$$\mathbb{E}(\lambda_{ij}) = 1 - (1-q)^2 = q(2-q) \le 2q = \frac{2c_1}{\sqrt{\log n}}.$$

Next, we investigate the concentration bound for $\lambda_{ij}$ and then for $\lambda = \max_{ij} \lambda_{ij}$. We recall the following one-sided Chernoff bound [4] for independent Bernoulli random variables $\{X_l\}_{l=1}^n$ with means $\{p_l\}_{l=1}^n$, $\bar{p} = \sum_{l=1}^n p_l/n$, and any $\eta \ge 1$:

$$\Pr\left(\frac{1}{n}\sum_{l=1}^n X_l > (1+\eta)\bar{p}\right) < e^{-\frac{\eta^2}{2+\eta}\bar{p}n}. \tag{32}$$

Applying (32) with the random variables $X_k = \mathbf{1}_{\{k \in B_{ij}\}}$ and $\eta = 1$,

$$\Pr\left(\lambda_{ij} > \frac{4c_1}{\sqrt{\log n}}\right) < e^{-\frac{2c_1}{3}\frac{|N_{ij}|}{\log |N_{ij}|}}. \tag{33}$$

To control the size of $|N_{ij}|$ in above probability bound, we use the following Chernoff bound [4] for i.i.d. Bernoulli random variables $\{X_l\}_{l=1}^m$ with means $\mu$ and any $0 < \eta < 1$:

$$\Pr\left(\left|\frac{1}{m}\sum_{l=1}^m X_l - \mu\right| > \eta\mu\right) < 2e^{-\frac{\eta^2}{3}\mu m}. \tag{34}$$

We note that by applying (34) with the random variables $\mathbf{1}_{\{k \in N_{ij}\}}$ and $\eta = 1/2$, we obtain that

$$\Pr\left(N_{ij} < \frac{1}{2}np^2\right) < 2e^{-\frac{1}{12}np^2}. \tag{35}$$

By combining the bounds in (33) and (35), we have for sufficiently large $n$

$$\Pr\left(\lambda_{ij} > \frac{4c_1}{\sqrt{\log n}}\right) < e^{-\frac{2c_1}{3}\frac{\frac{1}{2}np^2}{\log \frac{1}{2}np^2}} + 2e^{-\frac{1}{12}np^2} < e^{-\frac{c_1}{4}np^2} + 2e^{-\frac{1}{12}np^2}. \tag{36}$$

By applying a union bound over $ij \in E$, we have

$$\Pr\left(\lambda > \frac{4c_1}{\sqrt{\log n}}\right) < |E|e^{-\frac{c_1}{4}np^2} + 2|E|e^{-\frac{1}{12}np^2}, \tag{37}$$

where $\lambda = \max_{ij} \lambda_{ij}$. Therefore, with $q = c_1/\sqrt{\log n}$ and sufficiently large $n$, we have

$$\lambda < \frac{4c_1}{\sqrt{\log n}} \quad \text{w.p.} \quad 1 - |E|e^{-\frac{c_1}{4}np^2} - 2|E|e^{-\frac{1}{12}np^2} \tag{38}$$

Finally, we show for a proper constant $c_1$, (31) holds with high probability, and the exact recovery is concluded. We note that the RHS of (31) is lower bounded by

$$1 + \frac{eC_\alpha}{\mu} - \sqrt{\frac{eC_\alpha}{\mu}\left(2 + \frac{eC_\alpha}{\mu}\right)} = \frac{1}{1 + \frac{eC_\alpha}{\mu} + \sqrt{\frac{eC_\alpha}{\mu}\left(2 + \frac{eC_\alpha}{\mu}\right)}}$$

$$\ge \frac{1}{2\sqrt{\frac{eC_\alpha}{\mu}\left(2 + \frac{eC_\alpha}{\mu}\right)}} > \frac{1}{2\left(2 + \frac{eC_\alpha}{\mu}\right)} > \frac{1}{\frac{3eC_\alpha}{\mu}} = \frac{\mu}{3eC_\alpha}. \tag{39}$$

Combining this estimate of $\mu$ with (39), we obtain

$$1 + \frac{eC_\alpha}{\mu} - \sqrt{\frac{eC_\alpha}{\mu}\left(2 + \frac{eC_\alpha}{\mu}\right)} > \frac{c}{3eC_\alpha}\frac{1}{\sqrt{\log n}}. \tag{40}$$

Therefore, to guarantee (31) it suffices to let RHS of (38) be bounded from above by the RHS of (40). Namely, we require that

$$\frac{4c_1}{\sqrt{\log n}} < \frac{c}{3eC_\alpha}\frac{1}{\sqrt{\log n}},$$

which can be easily satisfied by setting $c_1 < \frac{c}{12eC_\alpha}$. Therefore, with the order of $q = (c/(12eC_\alpha\sqrt{\log n}))$ and equivalently $\epsilon_b = (cp/(12eC_\alpha\sqrt{\log n}))$, we can guarantee (31) and hence exact recovery with the probability specified in (38). Consequently, we verify that (30) is sufficient for exact recovery with the latter probability.

We finally note that assuming (28), that is, $p \geq c_0 n^{-1/2} \log^{1/2} n$ for sufficiently large constant $c_0$, the probability specified in (38) is high. Indeed,

$$1 - |E|e^{-\frac{c_1}{4}np^2} - 2|E|e^{-\frac{1}{12}np^2} > 1 - 3n^2 \exp(-\min\{\frac{c_1}{4}, \frac{1}{12}\}np^2) \tag{41}$$

$$> 1 - 3n^2 \exp(-\min\{\frac{c_1}{4}, \frac{1}{12}\}c_0 \log n) = 1 - 3n^{2-\min\{\frac{c_1}{4}, \frac{1}{12}\}c_0}. \tag{42}$$

Thus, with high probability, the recovery conditions in Theorem 2.1 are satisfied when (28) and (29) hold. We thus verify the bounds reported for T-AAB in Table 1.

## C  Visualization of 3D sparse models on ETH3D

We compare 3D sparse point cloud reconstructions of Cycle-Sync, GLOMAP and Theia. For GLOMAP and Theia, we use their default reconstruction parameters. For the Cycle-Sync pipeline, we feed the resulting camera poses to the 3D point triangulator in COLMAP (it uses bundle adjustment for the triangulations, while fixing camera pose estimators) and return the sparse 3D model. The latter was done quickly without careful tuning of parameters. Table 2 compares the number of triangulated 3D points for different SfM pipelines and some 3D sparse models on ETH3D. Tables 3 and 4 demonstrate the actual 3D sparse models by these methods.

We observe from Table 2 that for 9 out of the 13 datasets, Cycle-Sync improves the number of triangulated 3D points. This leads to an improvement of the overall quality of reconstruction for these datasets as noticed in Tables 3 and 4. This is due to the improvement on initial camera poses thanks to Cycle-Sync. For the other 4 datasets, Cycle-Sync fails to recover a meaningful 3D sparse model. For two of these datasets (meadow and office) all three methods are not performing well. For one dataset (relief) Theia is the only one which performs well and for the last dataset (relief_2) GLOMAP performs better than the other two methods.

| Dataset | Cycle-Sync | GLOMAP | Theia |
|---|---|---|---|
| courtyard | **30851** | 27674 | 17835 |
| delivery_area | **49306** | 25403 | 9534 |
| electro | **28061** | 24477 | 2641 |
| facade | **86111** | 66302 | 70571 |
| kicker | **26649** | 18685 | 10232 |
| meadow | 647 | **667** | 649 |
| office | 1351 | **2686** | 1395 |
| pipes | **8662** | 3551 | 1423 |
| playground | **16885** | 11816 | 416 |
| relief | 1642 | 12617 | **29588** |
| relief_2 | 1695 | **16902** | 3212 |
| terrace | **25285** | 13898 | 7216 |
| terrains | **47485** | 25082 | 12876 |

Table 2: Number of triangulated 3D points for each dataset using Cycle-Sync, GLOMAP, and Theia

| Dataset | Cycle-Sync | GLOMAP | Theia |
|---|---|---|---|
| courtyard | | | |
| delivery_area | | | |
| electro | | | |
| facade | | | |
| kicker | | | |
| meadow | | | |
| office | | | |
| pipes | | | |
| playground | | | |

Table 3: Triangulated 3D reconstructions (Part 1) using Cycle-Sync, GLOMAP, and Theia.

## D  Visualization of Camera Pose Estimation

We demonstrate pose estimation results for four scenes: *courtyard*, *meadow*, *office*, and *pipes*. For each scene, we compare the ground-truth camera poses with those estimated by GLOMAP, Theia, and Cycle-Sync. Figures 5–8 illustrate these comparisons. For the first three scenes, Cycle-Sync produces camera poses that align more closely with the ground truth, while GLOMAP and Theia exhibit larger misalignments. In the *pipes* scene, both Cycle-Sync and GLOMAP achieve good alignment, whereas Theia fails to produce meaningful results.

| Dataset | Cycle-Sync | GLOMAP | Theia |
|---|---|---|---|
| relief | | | |
| relief_2 | | | |
| terrace | | | |
| terrains | | | |

Table 4: Triangulated 3D reconstructions (Part 2) using Cycle-Sync, GLOMAP, and Theia.

# E Supplementary Tables for ETH3D

We provide additional tables and figures to demonstrate the pose estimation quality of Cycle-Sync. Table 5 demonstrates the location error for each SfM pipeline. Table 6 demonstrates the location error for different location estimation algorithms. Table 7 demonstrates the effect of each building block of Cycle-Sync by beginning with the LUD pipeline, and gradually adding MPLS, MPLS-cycle, Cycle-Sync and STE.

Table 5: Translation Error of each SfM pipeline on ETH3D. Here $\bar{t}$ and $\hat{t}$ denote the mean translation error and median translation error, respectively. BA refers to bundle adjustment.

| Scene | Cycle-Sync | | LUD | | GLOMAP (with BA) | | Theia (with BA) | |
|---|---|---|---|---|---|---|---|---|
| | $\bar{t}$ | $\hat{t}$ | $\bar{t}$ | $\hat{t}$ | $\bar{t}$ | $\hat{t}$ | $\bar{t}$ | $\hat{t}$ |
| courtyard | 0.27 | 0.02 | 0.85 | 0.75 | 0.34 | 0.03 | 0.01 | 0.01 |
| delivery_area | 0.15 | 0.04 | 0.37 | 0.24 | 0.01 | 0.00 | 0.09 | 0.00 |
| electro | 0.21 | 0.03 | 0.30 | 0.10 | 0.01 | 0.01 | 0.01 | 0.01 |
| facade | 0.25 | 0.00 | 0.43 | 0.18 | 1.01 | 1.01 | 0.01 | 0.00 |
| kicker | 0.02 | 0.01 | 0.09 | 0.02 | 0.36 | 0.01 | 0.01 | 0.01 |
| meadow | 0.02 | 0.02 | 0.40 | 0.28 | 0.91 | 1.01 | 0.68 | 0.51 |
| office | 0.20 | 0.03 | 0.17 | 0.03 | 0.06 | 0.01 | 0.95 | 0.97 |
| pipes | 0.01 | 0.01 | 0.06 | 0.03 | 0.01 | 0.01 | 0.01 | 0.00 |
| playground | 0.12 | 0.01 | 0.40 | 0.13 | 1.30 | 0.99 | 0.01 | 0.00 |
| relief | 0.00 | 0.00 | 0.00 | 0.00 | 0.90 | 0.78 | 0.01 | 0.01 |
| relief_2 | 0.01 | 0.01 | 0.70 | 0.73 | 0.01 | 0.01 | 0.81 | 0.79 |
| terrace | 0.01 | 0.01 | 0.01 | 0.01 | 0.01 | 0.01 | 0.01 | 0.01 |
| terrains | 0.01 | 0.01 | 0.02 | 0.01 | 0.00 | 0.00 | 0.42 | 0.05 |
| **Average** | **0.10** | **0.01** | 0.29 | 0.19 | 0.38 | 0.30 | 0.23 | 0.18 |

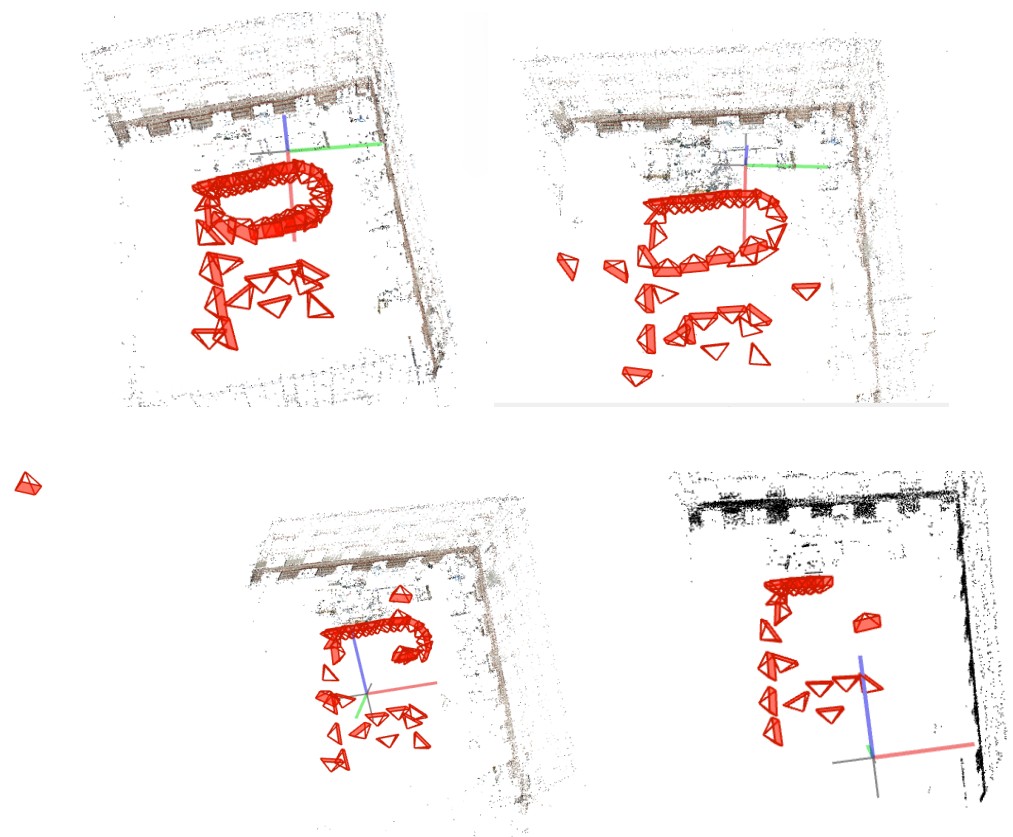

Figure 5: Visualization of camera location estimations and ground truth on the courtyard dataset. Top left: ground truth. Top right: Cycle-Sync. Bottom left: GLOMAP. Bottom right: Theia.

| Dataset | LUD | | BATA | | ShapeFit | | FusedTA | | Cycle-Sync | |
|---|---|---|---|---|---|---|---|---|---|---|
| | $\bar{t}$ | $\hat{t}$ | $\bar{t}$ | $\hat{t}$ | $\bar{t}$ | $\hat{t}$ | $\bar{t}$ | $\hat{t}$ | $\bar{t}$ | $\hat{t}$ |
| courtyard | 0.37 | 0.02 | 0.41 | 0.08 | 0.32 | 0.02 | 0.21 | 0.05 | 0.27 | 0.02 |
| delivery area | 0.25 | 0.16 | 0.35 | 0.23 | 0.18 | 0.08 | 0.96 | 0.93 | 0.15 | 0.04 |
| electro | 0.24 | 0.06 | 0.23 | 0.06 | 0.21 | 0.02 | 0.22 | 0.04 | 0.21 | 0.03 |
| facade | 0.30 | 0.01 | 0.28 | 0.01 | 0.98 | 0.98 | 0.91 | 0.86 | 0.25 | 0.00 |
| kicker | 0.03 | 0.01 | 0.03 | 0.01 | 0.02 | 0.01 | 0.03 | 0.02 | 0.02 | 0.01 |
| meadow | 0.06 | 0.02 | 0.11 | 0.05 | 0.08 | 0.02 | 0.08 | 0.02 | 0.02 | 0.02 |
| office | 0.20 | 0.03 | 0.21 | 0.03 | 0.20 | 0.03 | 0.21 | 0.03 | 0.20 | 0.03 |
| pipes | 0.01 | 0.01 | 0.02 | 0.01 | 0.01 | 0.01 | 0.01 | 0.01 | 0.01 | 0.01 |
| playground | 0.16 | 0.03 | 0.15 | 0.04 | 0.08 | 0.01 | 0.17 | 0.08 | 0.12 | 0.01 |
| relief | 0.00 | 0.00 | 0.03 | 0.01 | 0.00 | 0.00 | 0.00 | 0.00 | 0.00 | 0.00 |
| relief 2 | 0.18 | 0.19 | 0.04 | 0.03 | 0.90 | 0.90 | 0.11 | 0.08 | 0.01 | 0.01 |
| terrace | 0.01 | 0.01 | 0.01 | 0.01 | 0.01 | 0.01 | 0.01 | 0.01 | 0.01 | 0.01 |
| terrains | 0.01 | 0.01 | 0.06 | 0.06 | 0.01 | 0.00 | 0.02 | 0.01 | 0.01 | 0.01 |
| **Average** | 0.14 | 0.04 | 0.15 | 0.05 | 0.23 | 0.16 | 0.23 | 0.16 | **0.10** | **0.01** |

Table 6: Comparison of mean ($\bar{t}$) and median ($\hat{t}$) translation error for each ETH3D scene for different location estimation algorithms.

## F   Runtime

Table 8 compares the runtime of location estimation methods on ETH3D. We observe that STE-based methods are significantly faster than non-STE methods, including LUD+IRLS (the old LUD pipeline). In particular, Cycle-Sync runtime is $48\%$ lower than that of the LUD pipeline. Although Cycle-Sync is slower than common location estimation algorithms such as BATA, ShapeFit, and FusedTA, its

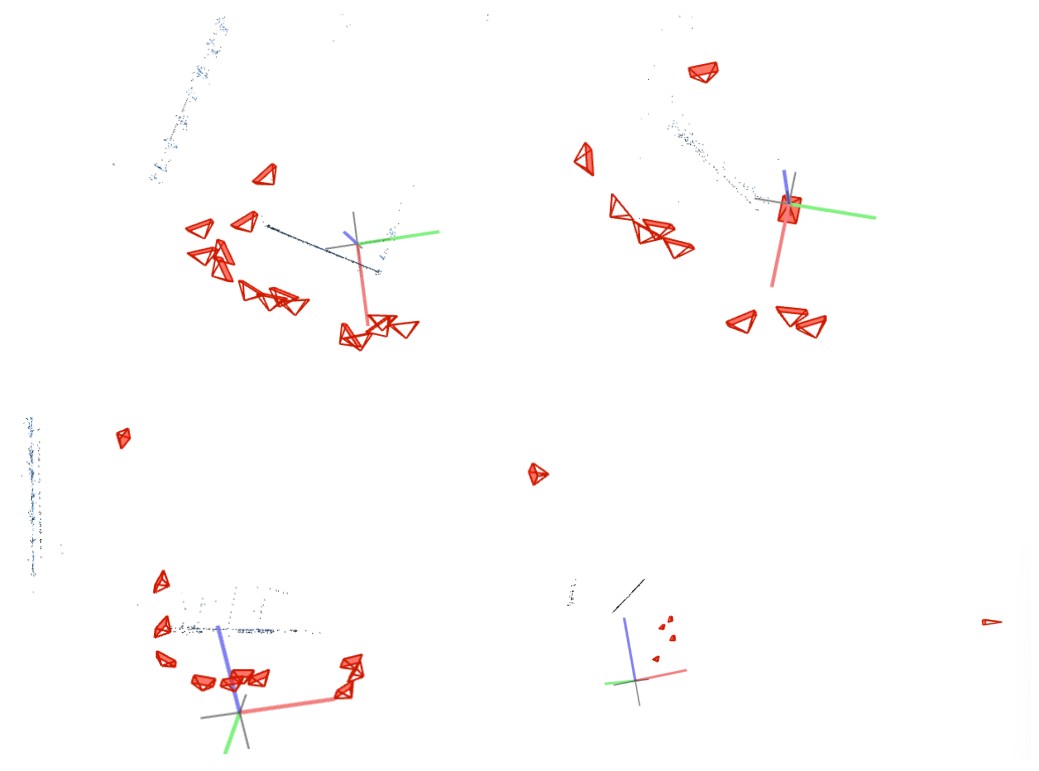

Figure 6: Visualization of camera location estimations and ground truth on the meadow dataset. Top left: ground truth. Top right: Cycle-Sync. Bottom left: GLOMAP. Bottom right: Theia.

| Scene | LUD+IRLS | | LUD+MPLS | | LUD+MPLS-cycle | | Cycle-Sync+MPLS-cycle | | STE+Cycle-Sync+MPLS-cycle | |
|---|---|---|---|---|---|---|---|---|---|---|
| | $\bar{t}$ | $\hat{t}$ | $\bar{t}$ | $\hat{t}$ | $\bar{t}$ | $\hat{t}$ | $\bar{t}$ | $\hat{t}$ | $\bar{t}$ | $\hat{t}$ |
| courtyard | 0.85 | 0.75 | 0.80 | 0.71 | 0.82 | 0.78 | 0.76 | 0.39 | 0.27 | 0.02 |
| delivery area | 0.37 | 0.24 | 0.36 | 0.23 | 0.37 | 0.28 | 0.16 | 0.06 | 0.15 | 0.04 |
| electro | 0.30 | 0.10 | 0.31 | 0.11 | 0.31 | 0.10 | 0.24 | 0.04 | 0.21 | 0.03 |
| facade | 0.43 | 0.18 | 0.49 | 0.19 | 0.53 | 0.04 | 0.26 | 0.00 | 0.25 | 0.00 |
| kicker | 0.09 | 0.02 | 0.07 | 0.03 | 0.10 | 0.05 | 0.03 | 0.01 | 0.02 | 0.01 |
| meadow | 0.39 | 0.28 | 0.49 | 0.47 | 0.48 | 0.23 | 0.18 | 0.06 | 0.02 | 0.02 |
| office | 0.17 | 0.03 | 0.18 | 0.03 | 0.22 | 0.03 | 0.21 | 0.03 | 0.20 | 0.03 |
| pipes | 0.06 | 0.03 | 0.05 | 0.02 | 0.05 | 0.02 | 0.01 | 0.01 | 0.01 | 0.01 |
| playground | 0.40 | 0.13 | 0.36 | 0.12 | 0.42 | 0.19 | 0.23 | 0.01 | 0.12 | 0.01 |
| relief | 0.00 | 0.00 | 0.00 | 0.00 | 0.00 | 0.00 | 0.00 | 0.00 | 0.00 | 0.00 |
| relief 2 | 0.70 | 0.73 | 0.15 | 0.15 | 0.15 | 0.15 | 0.01 | 0.01 | 0.01 | 0.01 |
| terrace | 0.01 | 0.01 | 0.01 | 0.01 | 0.01 | 0.01 | 0.01 | 0.01 | 0.01 | 0.01 |
| terrains | 0.02 | 0.01 | 0.01 | 0.01 | 0.01 | 0.00 | 0.01 | 0.00 | 0.01 | 0.01 |
| **Average** | 0.29 | 0.19 | 0.25 | 0.16 | 0.27 | 0.14 | 0.16 | 0.05 | **0.10** | **0.01** |

Table 7: Translation errors ($\bar{t}$ = mean translation error, $\hat{t}$ = median translation error) across all methods for ablation study.

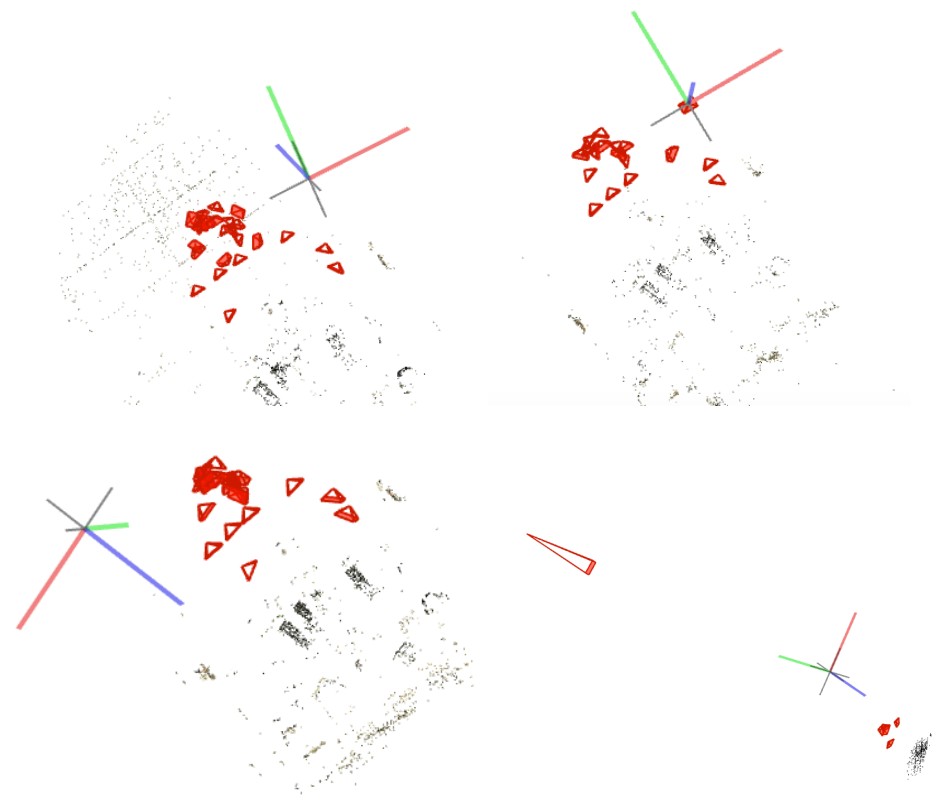

Figure 7: Visualization of camera location estimations and ground truth on the office dataset. Top left: ground truth. Top right: Cycle-Sync. Bottom left: GLOMAP. Bottom right: Theia.

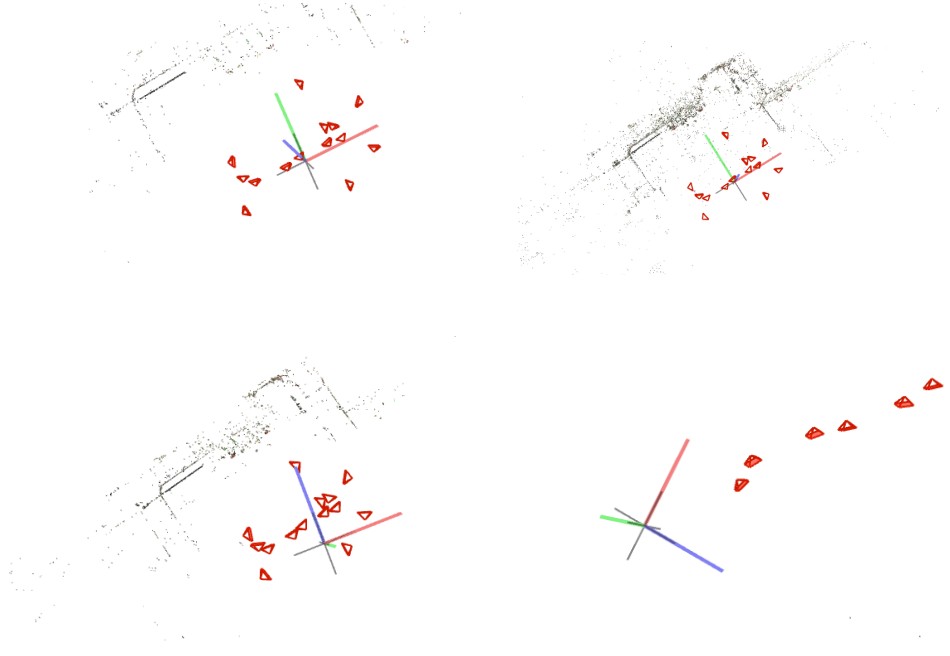

Figure 8: Visualization of camera location estimations and ground truth on the pipes dataset. Top left: ground truth. Top right: Cycle-Sync. Bottom left: GLOMAP. Bottom right: Theia.

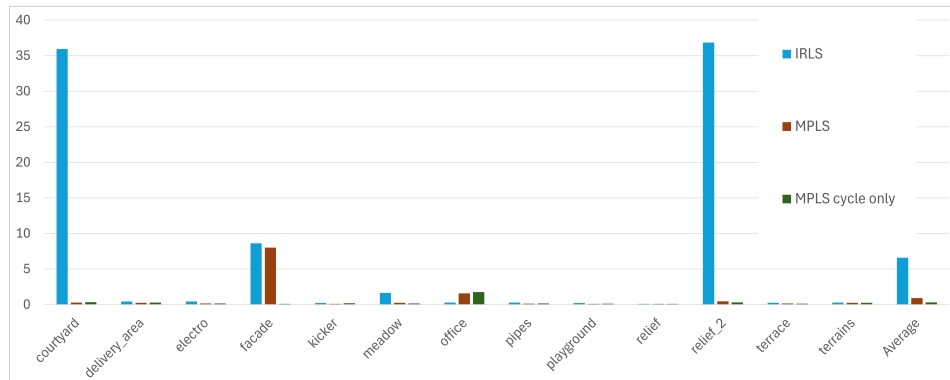

Figure 9: Rotation error (degrees) comparison on ETH3D for different rotation synchronization solvers.

runtime remains within the same order of magnitude, while achieving superior accuracy and stability in camera pose estimation.

| Scene | LUD+IRLS | LUD+MPLS | STE-based (with MPLS-cycle) | | | | |
|---|---|---|---|---|---|---|---|
| | | | STE+LUD | STE+BATA | STE+ShapeFit | STE+FusedTA | Cycle-Sync |
| courtyard | 23.60 | 19.38 | 6.10 | 5.83 | 5.74 | 6.01 | 10.05 |
| delivery area | 16.99 | 15.97 | 4.13 | 4.06 | 3.95 | 4.16 | 8.38 |
| electro | 15.58 | 13.92 | 4.21 | 3.92 | 3.82 | 4.19 | 8.58 |
| facade | 100.68 | 89.37 | 24.48 | 23.88 | 24.47 | 24.32 | 30.41 |
| kicker | 14.31 | 14.30 | 3.94 | 4.90 | 3.63 | 3.84 | 8.32 |
| meadow | 0.95 | 0.92 | 0.42 | 0.40 | 0.41 | 0.58 | 4.63 |
| office | 1.89 | 1.70 | 0.91 | 0.86 | 0.87 | 1.02 | 5.16 |
| pipes | 2.23 | 2.13 | 0.73 | 0.64 | 0.65 | 0.77 | 4.82 |
| playground | 10.78 | 9.60 | 2.79 | 2.62 | 2.57 | 2.84 | 7.34 |
| relief | 5.49 | 5.09 | 1.48 | 1.54 | 1.47 | 1.51 | 5.68 |
| relief 2 | 7.93 | 7.90 | 2.33 | 2.19 | 2.16 | 2.36 | 7.05 |
| terrace | 8.38 | 8.53 | 1.98 | 1.99 | 1.89 | 2.01 | 6.47 |
| terrains | 13.18 | 12.72 | 4.37 | 3.80 | 3.70 | 3.96 | 8.88 |
| **Average** | 17.08 | 15.50 | 4.45 | 4.36 | **4.26** | 4.43 | 8.91 |

Table 8: Runtime comparison (in seconds) of different SfM pipelines on ETH3D.

## G   Table and Figures for Rotation Synchronization

In this section we show the tables and figures for rotation errors. Figure 9 demonstrates the rotation errors on ETH3D across different rotation synchronization methods. Table 9 demonstrates the rotation errors on ETH3D across different pipelines.

We observe that MPLS-cycle (used in our Cycle-Sync) greatly improves rotation accuracy over existing pipelines. On average, MPLS-cycle reduces the median rotation error by $62.8\%$ and mean rotation error by $74.1\%$, compared to the best existing pipeline GLOMAP. Also, our proposed MPLS-cycle reduces the mean rotation error of MPLS by $56.6\%$ and median rotation error by $64.8\%$. It is worth noting that Cycle-Sync outperforms GLOMAP and Theia even without bundle adjustment. This demonstrates that even without bundle adjustment, our approach outperforms baselines that rely on it.

## H   Additional Experiment for IMC-PT

In this section we compare the camera location estimation results for different location estimation algorithms on IMC-PT. This dataset consists of 9 city-scale image sets, as well as ground truth camera poses estimated by aligning COLMAP SfM model with a LiDAR scan. We generate image matches using LoFTR [29], a deep learning feature matching method instead of SIFT. We use LoFTR since

Table 9: Comparison of rotation error (degrees) on ETH3D for different pipelines. Here $\bar{R}$, $\hat{R}$ means the mean rotation error and the median rotation error measured in degrees ($0°$-$180°$) respectively. BA refers to bundle adjustment.

| Scene | Cycle-Sync | | LUD | | GLOMAP (with BA) | | Theia (with BA) | |
|---|---|---|---|---|---|---|---|---|
| | $\bar{R}$ | $\hat{R}$ | $\bar{R}$ | $\hat{R}$ | $\bar{R}$ | $\hat{R}$ | $\bar{R}$ | $\hat{R}$ |
| courtyard | 0.78 | 0.36 | 43.85 | 35.93 | 3.50 | 1.61 | 0.14 | 0.10 |
| delivery_area | 0.76 | 0.28 | 0.78 | 0.43 | 0.09 | 0.07 | 0.41 | 0.21 |
| electro | 1.11 | 0.18 | 1.28 | 0.45 | 0.11 | 0.11 | 0.07 | 0.06 |
| facade | 0.18 | 0.11 | 18.10 | 8.64 | 0.78 | 0.35 | 0.09 | 0.09 |
| kicker | 0.44 | 0.21 | 0.25 | 0.22 | 0.47 | 0.22 | 0.11 | 0.09 |
| meadow | 0.29 | 0.18 | 2.94 | 1.64 | 21.13 | 6.05 | 3.14 | 3.37 |
| office | 3.39 | 1.69 | 8.98 | 0.29 | 0.69 | 0.34 | 7.23 | 6.34 |
| pipes | 0.17 | 0.18 | 0.34 | 0.28 | 0.12 | 0.13 | 0.08 | 0.09 |
| playground | 0.17 | 0.14 | 0.34 | 0.22 | 0.67 | 0.23 | 0.07 | 0.09 |
| relief | 0.11 | 0.11 | 0.11 | 0.12 | 3.18 | 1.68 | 0.14 | 0.15 |
| relief_2 | 0.29 | 0.31 | 41.78 | 36.84 | 0.10 | 0.10 | 36.85 | 34.23 |
| terrace | 0.15 | 0.13 | 0.27 | 0.27 | 0.12 | 0.12 | 0.15 | 0.13 |
| terrains | 0.22 | 0.25 | 0.27 | 0.29 | 0.19 | 0.21 | 3.54 | 1.78 |
| **Average** | **0.62** | **0.32** | 9.18 | 6.59 | 2.40 | 0.86 | 4.00 | 3.59 |

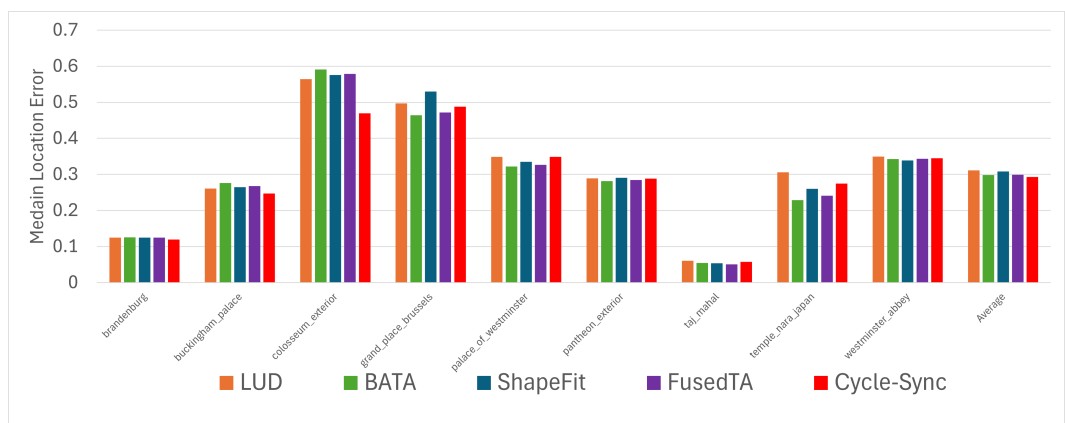

Figure 10: Median translation error for each IMC-PT scene and their average. The last column denotes the average median error across all datasets.

it is proved to be effective on popular homography estimation, relative pose estimation and visual localization benchmarks. Table 10 and Figure 10 demonstrate the location error of different location estimation algorithms, where rotation synchronization method is MPLS-cycle and all methods use STE.

We observe that Cycle-Sync achieves the smallest mean and median location error averaging on all datasets. For 3 out of 9 datasets, Cycle-Sync achieves both the smallest mean and median error. For other datasets, Cycle-Sync achieves no significantly larger mean and median error. The largest difference from Cycle-Sync to the best method in mean and median location error is 0.05, which is small compared to the average scale of location. The improvement of Cycle-Sync is smaller than that in ETH3D. We believe the reason is that LoFTR is not a good feature for this dataset, since it tends to overfit repetitive structures such as windows and facades. To sum up, while the gains over baselines are smaller than on ETH3D, Cycle-Sync still achieves the best mean and median across most scenes.

Table 10: Translation Errors ($\bar{t}$ and $\hat{t}$) for IMC-PT.

| Scene | LUD | | BATA | | ShapeFit | | FusedTA | | Cycle-Sync | |
|---|---|---|---|---|---|---|---|---|---|---|
| | $\bar{t}$ | $\hat{t}$ | $\bar{t}$ | $\hat{t}$ | $\bar{t}$ | $\hat{t}$ | $\bar{t}$ | $\hat{t}$ | $\bar{t}$ | $\hat{t}$ |
| brandenburg gate | 0.24 | 0.13 | 0.25 | 0.13 | 0.24 | 0.12 | 0.24 | 0.12 | 0.24 | 0.12 |
| buckingham palace | 0.36 | 0.26 | 0.37 | 0.28 | 0.38 | 0.26 | 0.37 | 0.27 | 0.35 | 0.25 |
| colosseum exterior | 0.75 | 0.56 | 0.90 | 0.59 | 0.67 | 0.58 | 0.92 | 0.58 | 0.56 | 0.47 |
| grand place brussels | 0.55 | 0.50 | 0.54 | 0.46 | 0.60 | 0.53 | 0.54 | 0.47 | 0.54 | 0.49 |
| palace of westminster | 0.43 | 0.35 | 0.40 | 0.32 | 0.44 | 0.33 | 0.41 | 0.33 | 0.44 | 0.35 |
| pantheon exterior | 0.44 | 0.29 | 0.44 | 0.28 | 0.44 | 0.29 | 0.44 | 0.28 | 0.44 | 0.29 |
| taj mahal | 0.12 | 0.06 | 0.12 | 0.05 | 0.12 | 0.05 | 0.12 | 0.05 | 0.12 | 0.06 |
| temple nara japan | 0.48 | 0.31 | 0.43 | 0.23 | 0.45 | 0.26 | 0.43 | 0.24 | 0.46 | 0.27 |
| westminster abbey | 0.87 | 0.35 | 0.84 | 0.34 | 0.83 | 0.34 | 0.84 | 0.34 | 0.89 | 0.35 |
| **Average** | 0.47 | 0.31 | 0.48 | 0.30 | 0.46 | 0.31 | 0.48 | 0.30 | **0.45** | **0.29** |

# I   Additional Supplementary Tables for ETH3D

## I.1   Sensitivity to Initialization

In tables 12 and 11 we report the mean and median location error on ETH3D data (averaged over different scenes) after several iterations for T-AAB and trivial initialization schemes. While the T-AAB initialization accelerates convergence by providing better starting weights, it does not significantly influence the final accuracy. Even trivial initialization using uniform weights performs similarly after sufficient iterations. Therefore, our method is robust even to trivial initialization, let alone variations in the T-AAB parameter.

| Iteration | Mean Error | Median Error |
|---|---|---|
| 5 | 0.110 | 0.020 |
| 10 | 0.100 | 0.017 |
| 15 | 0.099 | 0.016 |
| 20 | 0.099 | 0.016 |

Table 11: Performance with trivial (uniform) initialization.

| Iteration | Mean Error | Median Error |
|---|---|---|
| 5 | 0.102 | 0.018 |
| 10 | 0.099 | 0.016 |
| 15 | 0.099 | 0.015 |
| 20 | 0.099 | 0.014 |

Table 12: Performance with T-AAB initialization.

## I.2   Replacing LUD with BATA in Our Pipeline

In table 13 we show results of replacing LUD with BATA within our reweighting framework. Indeed, integrating LUD under Cycle-Sync's reweighting leads to better performance compared to integrating BATA. This is likely due to the stronger constraint of LUD for preventing collapsed trivial solution (the constraint is enforced on every edge). Moreover, our Welsh-type objective function already accounts

for large variations in distances, making angle-based methods such as BATA less advantageous in this case.

| Method | Mean Error | Median Error |
|---|---|---|
| Ours-LUD | 0.099 | 0.014 |
| Ours-BATA | 0.202 | 0.079 |

Table 13: Comparison between LUD and BATA integration under the Cycle-Sync framework on ETH3D data.

### I.3  Annealing Schedule $\lambda_t$

In tables 14 and 15 we include synthetic experiments for different annealing schedules $\lambda_t$ in settings with both additive noise and high corruption. Table 14 uses uniform corruption with $q = 0.7$ and higher noise level $\sigma = 0.2$.

| $\lambda_t$ | $\frac{t}{10+t}$ (ours) | 0 | $\frac{10}{10+t}$ | 1 | $\frac{t}{t+5}$ |
|---|---|---|---|---|---|
| median err | 0.24 | 0.36 | 0.27 | 0.37 | 0.29 |

Table 14: Uniform corruption $q = 0.7$, noise level $\sigma = 0.2$.

Table 15 the adversarial corruption with $q = 0.45$ (close to the theoretical limit $q = 0.5$) and higher noise level $\sigma = 0.2$.

| $\lambda_t$ | $\frac{t}{10+t}$ (ours) | 0 | $\frac{10}{10+t}$ | 1 | $\frac{t}{t+5}$ |
|---|---|---|---|---|---|
| median err | 0.17 | 0.32 | 0.18 | 0.20 | 0.17 |

Table 15: Adversarial corruption $q = 0.45$, noise level $\sigma = 0.2$.

We observe that our choice of $\lambda_t$ has the lowest median error for both settings. Therefore, our proposed schedule strikes a good balance between residual-driven updates early on and cycle-consistency emphasis later. This schedule has consistently outperformed alternatives, particularly in settings with both additive noise and high corruption.

We remark that using only the residual for reweighting (i.e., setting $\lambda_t = 0$, which corresponds to IRLS) often leads to significantly higher errors compared to cycle-based reweighting methods. The underlying issue is that, in noisy settings, some bad edges may coincidentally exhibit low residuals. When this occurs, the aggressive reweighting imposed by the Welsch objective can assign them disproportionately large weights, thereby amplifying their adverse impact. In contrast, our cycle-based reweighting effectively overcomes this limitation: it is extremely unlikely for a bad edge to exhibit a low average cycle inconsistency, unless all cycles it participates in are consistent—an event that is highly improbable for corrupted measurements. Overall, we find that our annealing strategy consistently achieves the best performance across most scenarios.

We also observe this trend in the context of rotation synchronization. Figure G illustrates that emphasizing cycle-consistency can significantly reduce orientation error. However, for rotation there is no need for annealing as it does not rely on distance estimation.

## J  Additional Experiment on 1DSfM Datasets

We report the mean and median location errors for 1DSfM dataset (averaged over different scenes), with all methods consistently preprocessed using STE and MPLS-cycle in table 16. Note that the ground truth camera poses in 1DSfM are generated by Bundler, which is considered outdated compared to modern tools like COLMAP (and even COLMAP may lack the accuracy of laser scans). We remark that this could be less reliable when benchmarking high-precision location solvers.

| Method | Mean Error | Median Error |
|--------|------------|--------------|
| Ours | 0.292 | 0.115 |
| LUD | 0.314 | 0.132 |
| BATA | 0.362 | 0.130 |
| ShapeFit | 0.670 | 0.410 |
| FusedTA | 0.330 | 0.130 |

Table 16: Performance comparison on the 1DSfM (photo tourism) datasets, averaged over multiple scenes. All methods are processed with STE and MPLS-cycle.

