# OpenReview forum: "Cycle-Sync: Robust Global Camera Pose Estimation through Enhanced Cycle-Consistent Synchronization"
_NeurIPS.cc/2025/Conference — NeurIPS 2025 spotlight_

### Official Review · Reviewer_e5bW · 2025-06-18

**Clarity:** 3
**Significance:** 4
**Originality:** 3
**Rating:** 5
**Confidence:** 3

**Summary:**

The paper proposes a method for camera location synchronization which is resistant to corrupted data which is present in real-world Structure-from-Motion (SfM) applications. By using redesigned Message Passing Least Squares (MLPS) to minimize a Welsch function the paper shows high robustness, great performance and solves the limitations of Least Unsquared Deviations (LUD) and BATA-type objectives. The paper also establishes a deterministic exact-recovery guarantee under adversarial corruption which outperforms all prior work under standard probabilistic model constraints. The framework is then extended to the rotation synchronization problem to reduce the orientation error. Lastly the proposed model in the paper removes any need for bundle adjustment which are computationally expensive but present in all referenced SfM pipelines.

**Questions:**

None.

**Ethical Concerns:**

["NO or VERY MINOR ethics concerns only"]

**Final Justification:**

The authors have addressed all issues from my review, why I keep the rating of 5 and propose to accept this contibution.

**Limitations:**

yes

**Paper Formatting Concerns:**

Figures 3 and 4 should be improved so that all values are readable also on a printed version of the paper, e.g. by different scaling or other measures.

**Quality:**

3

**Strengths And Weaknesses:**

1. Quality - The paper is well written with all claims being well supported and methods used being increments of previous works. The paper provides a clear overview of the method and how it is used and what its drawbacks are. The two main downsides to the quality is that in some places there are missing references for statements that are clear to the authors but might not be clear to a third party. The second downside is that some graphs (i.e. Fig 3, Fig 4) are a bit illegible due to the absolute differences of values being shown so a better representation of those would make a difference.
2. Clarity - The paper is clearly written with all theoretical setups providing assumptions and proofs. The introduction, related work and methodology provide clear overviews of what the paper proposes. The experimental setup is properly noted and reproducible.
3. Significance - The paper provides a method that consistently outperforms referenced methods in the task it was tested for. It advances the topic and provides insights into sensor synchronization, not only for location but for rotation and direction.
4. Originality - The paper builds upon previous methods' limitations and solves them in a novel way, providing valuable insights into the topic.

---

> ### Author Rebuttal · Authors · 2025-07-31
>
> We are very grateful for your thoughtful and insightful comments. Below, we address each of your concerns in detail.
>
> **1. missing references for statements**
>
> Thank you for pointing this out. We will go through the manuscript carefully and ensure that all statements—especially those not common knowledge—are properly cited with relevant references. We are committed to improving the clarity and completeness of our exposition.
>
>
>
>
> **2. Legibility of Figures 3 and 4**
>
>
> We agree that the current figures can be made more informative. In the revision, we will normalize axes or apply log/relative scaling where appropriate to improve the legibility and ensure that important trends and differences are clearly visible.

---

> > ### Comment · Reviewer_e5bW · 2025-08-04
> >
> > The authors have addressed all issues from my review, why I keep the rating of 5 and propose to accept this contibution.

---

### Official Review · Reviewer_jMZZ · 2025-06-24

**Clarity:** 2
**Significance:** 2
**Originality:** 3
**Rating:** 4
**Confidence:** 4

**Summary:**

This paper proposes Cycle-Sync, a robust translation averaging method.

The key contributions are twofold:

1. A Welsch-type loss function, which is less sensitive to large variations of distances and preserves non-smoothness at the origin.
2. A message-passing least squares (MPLS) framework that integrates cycle-consistency information across 3-cycles.

Experiments on both synthetic and the ETH3D real datasets show that Cycle-Sync outperforms other translation averaging and SfM baselines.

**Questions:**

1. Add geometric illustrations or figures to intuitively explain what cycle consistency captures, and how it improves robustness in practice.
2. Clarify $d_{ij, k}$ in Algorithm 1. Are these precomputed from input directions or estimated during optimization?
3. Justify hyperparameter choices, especially the value of $a=4$ in the loss, other parameters, and the weight initialization. Including a ablation study would be helpful.
4. Compare with AAB [20] in the experiments. Since both methods rely on 3-cycle consistency, a direct comparison is necessary.
5. Consider evaluating on the 1DSfM dataset [26] to improve comparability with prior work.
6. Evaluate performance under higher noise (e.g., σ = 0.1 or higher) to further support the claim of robustness.
7. Clarify what is novel in the "plug-and-play outlier rejection module" beyond simply using STE.

Considering the above concerns, I give a borderline reject at this stage. However, I would like to increase my rating if the concerns are addressed.

**Ethical Concerns:**

["NO or VERY MINOR ethics concerns only"]

**Final Justification:**

The authors have addressed my most of the concerns during the rebuttal and follow-up discussion. However, I encourage more precise wording in the final version (e.g., since STE is applied rather than introduced, it should be described accordingly). I also recommend including the additional experiments and clarifications from the rebuttal to further strengthen the paper.

**Limitations:**

Yes.

**Quality:**

2

**Strengths And Weaknesses:**

**Strengths:**
1. Translation averaging is indeed an important component in SfM, and the paper tackles the challenging case of cycle-consistent corruption, where other methods underperform.
2. The proposed translation averaging fully exploits cycle consistency using message-passing least squares.
3. The Welsch-type loss provides both nonlinearity and robustness, offering advantages over L1/L2 losses.

**Weaknesses:**
1. The geometric intuition of cycle consistency is not clearly described. It would be helpful to Including intuitive geometric visualizations.
2. The choice of hyperparameters, such as setting the Welsch loss parameter $a=4$ in line 118, some parameters in line 169, and weight initialization as $w_{ij,0} = exp(-20\tilde{s}_{ij})$ in line 152, lacks sufficient justification or empirical support. Although the method is claimed to be insensitive to these choices, it would be beneficial to include an ablation study or comparative analysis to support this claim.
3. Fig. 2 is not self-explanatory and the propagation mechanism in the bottom-left part is unclear. It would be helpful to provide a more detailed textual description.
4. In Algorithm 1, it is not explained how the 3-cycle inconsistencies $d_{ij, k}_{k\inC_{ij}}$ are obtained or whether they are known a priori or estimated.
5. The plug-and-play outlier rejection module is claimed as a contribution, but it simply uses STE in line 215. While effective, the novelty may be overstated unless further integration or modification is described.
6. The paper only verifies the performance of real data on ETH3D dataset, while ShapeFit, BATA, Fused-TA, LUD and other methods include results on the widely used 1DSfM dataset [26]. It would be beneficial to provide the comparisons on that dataset.
7. In Fig. 3, the robustness is only demonstrated under mild noise $\sigma = 0.05$. It would be valuable to explore performance under higher noise levels to validate the claim of robustness.
8. The All-About-that-Base (AAB) [20] is discussed as a related work that also leverages 3-cycle consistency, but not included in the experiments. Since AAB is most similar in spirit to Cycle-Sync, a direct experimental comparison would be highly relevant.

---

> ### Author Rebuttal · Authors · 2025-07-31
>
> We are very grateful for your thoughtful and insightful comments. Below, we address each of your concerns in detail.
>
> **1. The geometric intuition of cycle consistency**
>
> Thank you for highlighting this. We will include a geometric visualization in the revision. Intuitively, cycle consistency implies that traversing the triangle formed by three cameras via their estimated displacements (distance $\times$ direction) should return to the starting point. Deviations from this indicate local inconsistency and are strong indicators of corruption.
>
>
> **2. The choice of parameters.**
>
> We observed that the method performs stably across a range of settings for $a$, $\beta$ and initialization, validating its robustness.
>
> Please refer to our response to Reviewer SY7H on Q2 about the sensitivity to $a$ and $\beta$.
>
> Please also see our response to Reviewer 37Zk on the comment #2 (sensitivity to initialization) and comment #5 (sensitivity to $\lambda_t$).
>
> **3. Fig. 2 is not self-explanatory.**
>
> We agree that Figure 2 can be made clearer. In the revised version, we will expand the caption to explain that the lower-left schematic illustrates how corruption scores are computed via a weighted average of cycle inconsistencies, with weights derived from residuals via an exponential decay function. This mechanism is central to our cycle-based reweighting strategy.
>
> **4. Explanation of $d_{ij, k}$ in Algorithm 1**
> This is a typo—it should be $\tilde d_{ij,k}$. These are not learned online but are precomputed using the AAB-inconsistency measure described just after line 156. They are then used to run the T-AAB procedure and initialize the edge weights. We will clarify this in the revision.
>
> **5. Novelty of STE-Based Filtering Module**
>
> Yes, we directly use STE to estimate direction vectors via the equation in line 210. However, to our knowledge, no prior work has used the number of inlier keypoint matches identified by STE—or any other robust subspace recovery (RSR) method—as a statistic for filtering out bad edges. We emphasize that this step (line 210) differs from traditional geometric verification using the epipolar constraint: our proposed statistic is directly tied to the estimated directions and is well-suited for identifying and rejecting bad ones.
>
> Thus, while STE itself is not novel, our use of it to construct a new filtering statistic is. In particular, although the LUD paper uses REAPER to estimate directions, they do not leverage inlier counts from REAPER—or any RSR method—as a basis for rejecting unreliable directions.
>
> **6. Experiments on the 1DSfM dataset.**
>
> We also answered this question in the reply to reviewer 37Zk which we copied here:
>
> In our earlier submission, we evaluated our method (we did not use STE and MPLS-cycle at the time) on the 1DSfM (photo tourism) datasets and achieved at least a 20% reduction in median error compared to the best baseline. However, reviewers raised concerns about our reliance on SIFT features and recommended switching to deep learning-based features. Another critique was that the ground truth camera poses in 1DSfM are generated by Bundler, which is considered outdated compared to modern tools like COLMAP—and even COLMAP may lack the accuracy of laser scans. This is particularly problematic when benchmarking high-precision location solvers. These limitations raise valid concerns about the suitability of older datasets for evaluating modern methods. Nevertheless, we report the mean and median location errors below (averaged over different 3D scenes), with all methods consistently preprocessed using STE and MPLS-cycle:
>
> | Method | mean error | median error |
> |:-------------------------:|:--------:|:----------:|
> |   Ours   |   0.292    |   0.115  |
> |   LUD   |    0.314    |   0.132  |
> |   BATA   |    0.362    |   0.130  |
> |   ShapeFit   |    0.670    |   0.410  |
> |   FusedTA   |    0.330    |   0.130  |
>
> For the aforementioned reasons, we conducted an additional experiment in the supplementary material on a city-scale dataset (an extended version of the original Photo Tourism dataset) using deep learning-based keypoint features. See Table 10 and Figure 11 in the supplementary material. However, the performance gain of Cycle-Sync on this dataset is noticeably smaller than that observed on ETH3D. We found that LoFTR is not well-suited for this dataset, as it tends to overfit repetitive structures such as windows and facades. Moreover, the ground truth camera poses are derived from COLMAP reconstructions, which we believe are significantly less accurate than those obtained from laser scans.
>
>
> **7.  performance under higher noise levels**
>
> We include additional experimental results evaluating Cycle-Sync and baselines under high additive noise $\sigma=0.2$ and high corruption. Our method continues to remain competitive.
>
> For example, for adversarial corruption when $q=0.45$ (close to the theoretical limit $q=0.5$) and  $\sigma=0.2$
>
> | Method | Ours | LUD | BATA| SF| FusedTA
> |:-------------------------:|:--------:|:----------:|:----------:|:----------:|:----------:|
> |   median error  |  0.17      |    0.91 |  0.31 | 1.47 | 0.23|
>
>
> For example, for uniform corruption when $q=0.7$ and  $\sigma=0.2$
>
>
> | Method | Ours | LUD | BATA| SF| FusedTA
> |:-------------------------:|:--------:|:----------:|:----------:|:----------:|:----------:|
> |   median error  |  0.24      |    0.63 |  0.27 | 1.44 | 0.52|
>
>
> We conclude by noting that in real datasets—assuming reasonably accurate keypoint features and ground truth camera poses—the error model is better characterized as corruption plus small noise. In practice, the relative measurements on edges are obtained via fundamental matrix estimation followed by RANSAC, which typically either succeed (correctly identify inlier keypoint match) or fail outright, with much less cases falling in between.
>
>
> **8. Comparison with AAB**
>
> We conducted an experiment on ETH3D where we removed edges flagged by AAB (based on its statistics), extracted the parallel rigid component, and ran LUD or other solvers on this cleaned graph. Before running AAB, the graph is already preprocessed by STE and MPLS-cycle for fair comparison. We report the mean and median location error averaged over different 3D scenes. While AAB-based preprocessing improves robustness to some extent, Cycle-Sync achieves much better results. This further demonstrates the advantage of our message-passing reweighting over standalone edge pruning.
>
> | Method | mean error | median error |
> |:-------------------------:|:--------:|:----------:|
> |   Ours   |   0.099    |   0.014  |
> |   AAB, remove 30% edges   |   0.153    |   0.092  |
>
>
> Removing 30% edges keeps 95% of the nodes. We note that our method is evaluated on the full graph, and even though our error is significantly smaller than AAB evaluated on a cleaner graph. Interestingly, AAB+LUD performs slightly worse than LUD alone. A possible reason is that STE has already performed the role of filtering bad edges—arguably more accurately than AAB. Applying AAB afterward may discard additional good edges or nodes, creating sparse regions in the graph and reducing the overall stability of the algorithm.

---

> > ### Comment · Reviewer_jMZZ · 2025-08-03
> >
> > Thanks for the authors' detailed rebuttal and additional experiments. The clarifications have addressed almost all of my questions. That said, I would like to raise two follow-up concerns:
> >
> > 1. As noted by Reviewer 37Zk in W3, the ablation study in Figure 4 (bottom panel) shows that the full combination of STE + Cycle-Sync + MPLS-cycle achieves the best performance, even compared to Cycle-Sync + MPLS-cycle alone. This suggests that a significant portion of the improvement may be attributable to the inclusion of STE. Since the STE module itself follows [27] without major modification, this raises questions about the marginal contribution of the proposed cycle-consistency refinement, which is positioned as the main novelty of this work.
> >
> > 2. Regarding the reduced performance on the extended PhotoTourism dataset (with LoFTR features) raises the question of how well Cycle-Sync generalizes beyond carefully curated datasets like ETH3D. The authors attribute this to LoFTR overfitting repetitive structures, which is plausible, but it might also suggest that the proposed method is sensitive to keypoint quality or scene regularity.

---

> > > ### Author Response · Authors · 2025-08-04
> > > **Clarifying the Depth of Our Contributions (Part 1)**
> > >
> > > Dear Reviewer jMZZ,
> > >
> > > Thank you for your additional comments.
> > >
> > > **1. Regarding your first concern:**
> > >
> > > We respectfully disagree with the suggestion that the improvements achieved by Cycle-Sync are not significant. The gains are not only statistically meaningful — they are, in fact, **unprecedented** among translation averaging methods.
> > >
> > > In the **bottom panel of Figure 4**, **without any preprocessing**, Cycle-Sync reduces the median error of the best baseline (LUD) by **64%**. When STE is applied, the error is further reduced by another **80%**. These gains are large and consistent.
> > >
> > > In the **middle panel**, where all methods are already preprocessed by STE, Cycle-Sync still outperforms the best baseline by **75%** in median error. Achieving such dominant improvements even after strong preprocessing is highly unusual and, to our knowledge, has not been reported before in this literature.
> > >
> > > In **synthetic experiments**, which isolate the core location solvers without any preprocessing like STE, Cycle-Sync alone achieves estimation errors that are up to **five orders** of magnitude lower (i.e., \(10^5\) times smaller) than those of competing methods under heavy corruption. This is a clear, quantitative demonstration of a breakthrough in robustness and accuracy.
> > >
> > > We also want to emphasize that our use of **STE is not a simple plug-in**. Originally, STE was developed for robust subspace recovery and covariance estimation—not for outlier rejection. In this work, we introduce a **novel geometric statistic** (inspired by equation 210) that measures the agreement between keypoint matches and the estimated direction vectors.
> > >
> > > To the best of our knowledge, this is the first time any Robust Subspace Recovery (RSR) method has been used in this way for robust location estimation.
> > >
> > > For comparison, prior approaches to outlier filtering or reweighting have relied on:
> > > - Cycle-consistency (e.g., AAB),
> > > - Residuals from rotation estimation (as in the BATA pipeline), or
> > > - Heuristics based on keypoint match density (e.g., assuming denser matches are more likely to be inliers, as in [1]).
> > >
> > > However, none of the existing location estimation methods have leveraged the geometric consistency of keypoint matches via the equation in line 210, nor have they attempted to evaluate the correctness of direction vectors using any Robust Subspace Recovery (RSR) method, such as STE.
> > > Our contribution goes beyond merely incorporating STE; we repurpose it to define a new, principled statistic tailored for robust geometric reasoning.

---

> > > > ### Author Response · Authors · 2025-08-04
> > > > **Clarifying the Depth of Our Contributions (Part 2)**
> > > >
> > > > **2. Regarding your second concern**
> > > >
> > > > As we mentioned, the old *Photo Tourism* dataset and its extensions—such as the newer variant used primarily for the Image Matching Challenge—are not ideal for evaluating high-precision location estimators. The newer version is tailored more toward image matching tasks rather than accurate camera pose estimation.
> > > >
> > > > In contrast, the **ETH3D** dataset is now widely regarded as a more reliable benchmark for evaluating Structure-from-Motion (SfM) methods, due to its high-quality laser-scan-based ground truth. Importantly, ETH3D remains challenging for many leading methods, as evident from the failure cases of GLOMAP and Theia shown in our supplementary material. Our results on ETH3D are comparable to GLOMAP, which itself reported performance on par with COLMAP. However, we emphasize that the *Photo Tourism* dataset uses COLMAP to *generate* the ground truth. This raises serious concerns about its validity for benchmarking new methods—after all, it is fundamentally problematic to use a baseline method as the source of ground truth. For this reason, we chose to include those results only in the supplementary material, and we caution against drawing conclusions from such evaluations.
> > > >
> > > > Despite the questionable ground truth in the **1DSfM** dataset, our results still demonstrate clear improvements. Using STE preprocessing for all baselines, our method achieves a **13%** reduction in median error over the best baseline. In contrast, other methods (excluding ShapeFit, which performs poorly) only vary within **±1%** of their average median error. This highlights the significance of our improvement.
> > > >
> > > > As for the **new Photo Tourism** dataset derived from the IMC benchmark—which is designed primarily for image matching—all methods perform similarly. This suggests that the bottleneck may lie in the dataset itself or its ground truth. Even so, our method reduces the median error by **3.3%** compared to the best baseline, while other methods vary within **±1.7%** of their mean performance—again showing that our gains are non-trivial.
> > > >
> > > > Among all datasets we evaluated, **ETH3D** provides the most accurate and trustworthy ground truth. Notably, on ETH3D, our method consistently outperforms all baselines *even without the STE module*—a level of robustness and accuracy not previously reported by other approaches.
> > > >
> > > > We would finally argue that, even with perfect ground truth, these concerns apply equally to all baseline methods. No single approach can perform optimally across all scenarios. For instance, in the case of pure Gaussian noise, a simple least squares method would outperform all others.

---

> > > > > ### Author Response · Authors · 2025-08-04
> > > > > **Clarifying the Depth of Our Contributions (Part 3)**
> > > > >
> > > > > **Substantive Theoretical Contribution Was Overlooked**
> > > > >
> > > > > Finally, we would like to respectfully highlight that the review appears to have overlooked one of the central contributions of our work: a **major theoretical breakthrough** in the field of location estimation. Independent of any algorithmic or empirical improvements (Cycle-Sync, STE), our paper makes the first significant progress in nearly 7 years on the sample complexity required for exact recovery in the presence of outliers.
> > > > >
> > > > > This question—how many correct/incorrect measurements are needed to provably recover all locations—has remained largely unresolved since the landmark result by Hand, Lee, and Voroninski, published in Communications on Pure and Applied Mathematics (CPAM), a top-tier journal in applied mathematics. That work required a sophisticated and technically demanding analysis, and no substantial improvement has been reported since—except a known bound for LUD, which we summarize in Table 1 of our paper.
> > > > >
> > > > > In this context, our work fundamentally advances the state of the art, offering the sharpest known phase transition guarantees for exact location recovery and significantly improving upon prior results. This represents a major step toward answering the ultimate question: What is the best possible phase transition bound for a location recovery algorithm? Among all existing works, ours comes closest to resolving this foundational question.

---

> > > > > > ### Author Response · Authors · 2025-08-04
> > > > > > **Reference**
> > > > > >
> > > > > > [1] It Is All in the Weights: Robust Rotation Averaging Revisited, 3DV 2021

---

> > > > > > > ### Author Response · Authors · 2025-08-04
> > > > > > > **Additional Clarification**
> > > > > > >
> > > > > > > We are unsure what led to the conclusion that our improvements are not significant. To avoid any potential confusion/misunderstanding, we would like to clarify the following:
> > > > > > >
> > > > > > >
> > > > > > > When we refer to combinations such as **STE + Cycle-Sync** or **Cycle-Sync + MPLS-cycle**, *Cycle-Sync* specifically denotes the **plain location solver** (Algorithm 1), without any preprocessing (e.g., STE).
> > > > > > >
> > > > > > >
> > > > > > > In contrast, when we refer to *Cycle-Sync* alone in the context of **real data**, it represents the **full pipeline**, including preprocessing steps. We did clarify these distinctions in our description of the experimental setup, but we would like to emphasize them again here in case they were overlooked:
> > > > > > >
> > > > > > >
> > > > > > > - **Figure 4, top panel**: *Cycle-Sync* refers to the **full pipeline**.
> > > > > > > - **Figure 4, middle panel**: *Cycle-Sync* again denotes the **full pipeline**; all other baselines are also processed with STE and MPLS-cycle. In this setting, we are strictly evaluating the location estimators, with all other components held fixed—thus isolating the effect of the location solver itself and **excluding the influence of STE and MPLS-cycle**.
> > > > > > > - **Figure 4, bottom panel**: *Cycle-Sync* refers to the **plain location solver** (Algorithm 1), without any preprocessing such as STE.
> > > > > > >
> > > > > > >
> > > > > > > For synthetic data, there shouldn’t be any ambiguity – *Cycle-Sync* denotes the **plain location solver** (Algorithm 1), without any preprocessing (e.g., STE).

---

> > > > > > > > ### Comment · Reviewer_jMZZ · 2025-08-08
> > > > > > > >
> > > > > > > > Thanks for the authors' detailed clarifications regarding the contributions. I still believe it would strengthen the final version to more clearly disentangle the improvements from the STE-based preprocessing versus the core Cycle-Sync location solver.
> > > > > > > >
> > > > > > > > 1. Regarding the use of STE, I remain unconvinced by the claim that it is not a simple plug-in. As acknowledged, STE was originally developed for robust subspace recovery [27], and its robustness to outliers makes it a natural fit for outlier rejection. While the proposed statistic based on keypoint agreement is interesting, I see this as a thoughtful reuse of an existing robust method, rather than a fundamentally new framework.
> > > > > > > >
> > > > > > > > 2. On the theoretical contribution, while the improved sample complexity results are valuable, I suggest being slightly more measured in referring to this as a “major theoretical breakthrough.” Recent work such as [1] also proposes a robust method suitable for translation averaging, and while the techniques may differ, it is also a strong contribution to this line of work.
> > > > > > > >
> > > > > > > > [1] Chitturi, Sidhartha, and Venu Madhav Govindu. "Adaptive annealing for robust averaging." European Conference on Computer Vision, 2024.

---

> > > > > > > > > ### Author Response · Authors · 2025-08-08
> > > > > > > > >
> > > > > > > > > We appreciate the reviewer’s follow-up, but we are concerned that the **main** points of our earlier rebuttal were not understood.
> > > > > > > > >
> > > > > > > > > **1. Disentangling STE from Cycle-Sync**
> > > > > > > > >
> > > > > > > > > Our results already make it clear that Cycle-Sync, **both with and without** any STE preprocessing, delivers unprecedented improvements over all baselines. This directly demonstrates the strength of Cycle-Sync itself and addresses the question of its standalone contribution. We repeat that in the bottom panel of Figure 4, the plain Cycle-Sync solver reduces the median error of the best baseline by **64%**—a margin that is both large and consistent across settings. Furthermore, with STE the improvement by Cycle-Sync is **75%** over other baselines with STE.
> > > > > > > > >
> > > > > > > > > Given these results, it is not clear why the emphasis remains so strongly on STE. Nevertheless, STE is an additional natural, modular enhancement that further improves robustness in practice, and it is unclear why we are criticized for adding this new component. Our paper gives full credit to the authors who invented STE, but applies STE to camera location estimation, so it is unclear why we are criticized for this. We believe authors should not be discouraged from including such additional results, directly inspired by other papers, as they offer valuable guidance to practitioners. Let us try to explain why we believe our application of STE (or any other robust subspace recovery method) has an original component. The point is that we further used it to infer bad direction vectors and exclude them and not just for finding a robust subspace (see lines 215-216 in the submitted manuscript). This strategy helped in improving performance. Nevertheless, this is a rather minor issue and we believe we have addressed the major issues.
> > > > > > > > >
> > > > > > > > > **2. On the theoretical contribution**
> > > > > > > > >
> > > > > > > > > We also stand by our description of the theoretical results as a major advancement. The long-standing open question in this area is: *What is the sharpest possible phase transition bound for exact location recovery under adversarial corruption?* Our work substantially improves upon the best-known bounds for ShapeFit and LUD—benchmarks that have remained unchallenged for nearly seven years.
> > > > > > > > >
> > > > > > > > > The ECCV 2024 paper that you mentioned is indeed a solid contribution, but it does **not** compete on this front: It focuses on algorithmic design, not on tightening fundamental sample complexity guarantees. In contrast, our analysis directly advances the theoretical frontier in a problem where such progress has been rare and technically challenging. We will clarify this distinction in the revision.

---

### Official Review · Reviewer_SY7H · 2025-07-01

**Clarity:** 4
**Significance:** 3
**Originality:** 4
**Rating:** 5
**Confidence:** 4

**Summary:**

Cycle-Sync is a global camera pose estimation method based on message-passing least squares for cycle consistency and Welsch-like robust loss, instead of L1 or L2 loss, for camera location estimation. With unknown inter-camera distance, Cycle-Sync adapts MPLS on translation estimation. Such a method avoids bundle adjustment in the classical Structure from Motion pipelines while showing superior performance.

**Questions:**

1. How do you define well-shaped triangles? How do you choose the lower and upper bounds used in truncated AAB (T-AAB)?.
2. Why does the proposed loss function use $\alpha$=4? How sensitive is the optimization w.r.t. the choice of $\alpha$? Or it can be any constant.
3. It seems the performance decreases a lot when removing the STE outlier filter in some scenes. Have you tried using STE for the baselines?
4. Others: see the weakness

**Ethical Concerns:**

["NO or VERY MINOR ethics concerns only"]

**Final Justification:**

Most of my concerns are addressed by the detailed rebuttal. I will keep my rating.

**Limitations:**

Yes

**Paper Formatting Concerns:**

None.

**Quality:**

4

**Strengths And Weaknesses:**

Strengths:
1. A novel global location estimation pipeline based on cycle consistency, showing decent improvement over the SOTA structure-from-motion methods with bundle adjustment.
2. A new optimization formulation, solved by cycle consistency, leading to reduced corruption level.
3. The paper is well-written and easy to read.

Weakness:
1. The contribution claimed to deal with the highly variant edges in distance. In principle, yes. However, there is only one real-world dataset,  ETH3D, used in the evaluation with its precise ground truth from the laser, which is not sufficient to conclude eliminating bundle adjustment for all.
2. Missing discussion. It is unclear how fast or how expensive the process of iteratively optimizing the cycle inconsistencies over the estimated locations is.

---

> ### Author Rebuttal · Authors · 2025-07-31
>
> We are very grateful for your thoughtful and insightful comments. Below, we address each of your concerns in detail.
>
> **1. other real-world dataset**
>
> We also answered this question in the reply to reviewer 37Zk which we copied here:
>
> In our earlier submission, we evaluated our method (we did not use STE and MPLS-cycle at the time) on the 1DSfM (photo tourism) datasets and achieved at least a 20% reduction in median error compared to the best baseline. However, reviewers raised concerns about our reliance on SIFT features and recommended switching to deep learning-based features. Another critique was that the ground truth camera poses in 1DSfM are generated by Bundler, which is considered outdated compared to modern tools like COLMAP—and even COLMAP may lack the accuracy of laser scans. This is particularly problematic when benchmarking high-precision location solvers. These limitations raise valid concerns about the suitability of older datasets for evaluating modern methods. Nevertheless, we report the mean and median location errors below (averaged over different scenes), with all methods consistently preprocessed using STE and MPLS-cycle:
>
> | Method | mean error | median error |
> |:-------------------------:|:--------:|:----------:|
> |   Ours   |   0.292    |   0.115  |
> |   LUD   |    0.314    |   0.132  |
> |   BATA   |    0.362    |   0.130  |
> |   ShapeFit   |    0.670    |   0.410  |
> |   FusedTA   |    0.330    |   0.130  |
>
> For the aforementioned reasons, we conducted an additional experiment in the supplementary material on a city-scale dataset (an extended version of the original Photo Tourism dataset) using deep learning-based keypoint features. See Table 10 and Figure 11 in the supplementary material. However, the performance gain of Cycle-Sync on this dataset is noticeably smaller than that observed on ETH3D. We found that LoFTR is not well-suited for this dataset, as it tends to overfit repetitive structures such as windows and facades. Moreover, the ground truth camera poses are derived from COLMAP reconstructions, which we believe are significantly less accurate than those obtained from laser scans.
>
>
>
> **2. Runtime Cost of Optimizing Cycle Inconsistencies**
>
> The additional cost of optimizing cycle inconsistencies can be directly inferred from Table 8 in the supplementary material. Specifically, the time difference between the STE+LUD and Cycle-Sync pipelines reflects the added cost of our message-passing-based optimization. We will clarify this in the main text and refer the reader to supplementary material for details.
>
> **Q1. Definition well-shaped triangles? The choice of the lower and upper bounds used in T-AAB?**
>
> We define well-shaped triangles as those that avoid extreme angle degeneracies. In particular, we require that all angles lie between $\alpha$ and $\pi-\alpha$ for some $\alpha \in (0, \pi/2)$. In practice we choose $\alpha = arcsin(0.6)\approx 37 degree$, based on empirical tuning on synthetic datasets. We observed similar performance for any $\alpha$ between arcsin(0.4) and arcsin(0.7). We note that a larger $\alpha$ often yields a better T-AAB estimate of corruption levels; however, $\alpha$ cannot be too large, as this would result in an insufficient number of cycles. We finally remark that our method is robust even to trivial initialization, let alone variations in the T-AAB parameter. Please see our reply to reviewer 37Zk on the comment #2 about the sensitivity to initialization.
>
> **Q2: Sensitivity to Parameters**
>
> We conducted an ablation study on the key parameters using ETH3D data: the constant $a$  in the robust loss function, the exponential decay constant $\beta$ in the cycle-consistency updates, and various initialization schemes. The mean and median errors are averaged across different 3D scenes. Our findings show that Cycle-Sync is not sensitive to these choices within reasonable ranges. For the sensitivity to $\lambda_t$ please see our reply to 37Zk on the comment #5.
>
> | Method | mean error | median error |
> |:-------------------------:|:--------:|:----------:|
> |   Ours   |   0.099    |   0.014  |
> |   $\beta = 4$   |   0.098    |   0.016  |
> |   $\beta = 8$   |   0.099    |   0.016  |
> |   $a = 2$   |   0.098    |   0.016  |
> |   $a= 8$   |   0.102    |   0.014  |
>
> **Q3: STE for the baselines?**
>
> This configuration is already included in the middle panel of Figure 4. As stated in lines 276–278, **all location solvers were preceded by the same STE** and MPLS-cycle preprocessing to ensure fairness. Without this common preprocessing, the performance gap between other baselines and Cycle-Sync would be even larger.

---

> > ### Comment · Reviewer_SY7H · 2025-08-05
> >
> > Thanks for the detailed rebuttal. Most of my concerns are addressed. It will be good to include experiments on both accurate laser GT (eg, ETH3D) and COLMAP ground truth, as it is widely used and built on a holdout image set.

---

> > > ### Author Response · Authors · 2025-08-06
> > > **Including both ground truths**
> > >
> > > Dear Reviewer SY7H,
> > >
> > > Thank you for carefully reviewing our rebuttal. As noted, we have included experiments on the IMC-extended PhotoTourism dataset (with COLMAP ground truth) in the supplementary material. We will also add the results on the original 1DSfM dataset (as reported in the rebuttal). If space permits, we will move these results back into the main paper.

---

### Official Review · Reviewer_37Zk · 2025-07-03

**Clarity:** 3
**Significance:** 3
**Originality:** 3
**Rating:** 5
**Confidence:** 4

**Summary:**

This paper proposes a framework for robust global location estimation of cameras, given pairwise direction estimates in the context of the Structure-from-Motion (SfM) problem. The approach formulates a LUD-type cost function using a robust Welsch loss and leverages distance-based cycle-consistency over cycles, where the distances are iteratively estimated. In the initial iteration, the weights are derived using angle-based cycle-consistency computed via the AAB method [20], which relies solely on the input relative directions.

To obtain more accurate global rotations during the rotation averaging step, the authors introduce a variant of the MPLS method that progressively prioritizes cycle-consistency over residual minimization. These improved global rotations are crucial for enhancing the quality of relative translation directions in the subsequent translation averaging stage.

Additionally, the authors propose a technique called STE, which robustly estimates the relative translation directions using a subspace recovery method [27]. A theoretical guarantee is provided, demonstrating that the proposed method can accurately recover the corruption levels of both inlier and outlier edges.

**Questions:**

1. When $\lambda_t$ increases from 0 to 1, the weighting shifts from residuals to cycle-inconsistency. While it’s understandable that cycle-inconsistency may be unreliable in early iterations due to dependence on the initial (and inaccurate) distance estimates, the residuals are also computed from the same initialization and are therefore equally unreliable. This raises a key question: what is the rationale for prioritizing residuals early on if they too are sensitive to initialization? I think that addressing weakness-5 may address this concern also.
2. It is unclear why the authors specifically advocate for the Welsch loss function. Other parametric robust loss functions, such as Geman-McClure or Huber, can exhibit similar behavior in handling variability in edge lengths when tuned appropriately. Can you correct me if this is wrong?
3. In MPLS vs MPLS-cycle for rotations averaging, what is meant by "disabling reweighting" in Lines 206-207?
4. Typo: In Algorithm-1, in the first two steps, should it be $\tilde{s_{ij}}$ instead of $s_{ij}$?
5. Although the 1DSfM datasets are older, they typically exhibit higher corruption levels than ETH3D. Given that the translation errors in Table 6 (supplementary) for ETH3D are already low, it’s unclear whether improvements in translation averaging significantly impact the final SfM output. As an optional evaluation, it would be valuable to test the method on more challenging datasets like 1DSfM to better highlight its robustness.

**Ethical Concerns:**

["NO or VERY MINOR ethics concerns only"]

**Final Justification:**

The authors extend the message-passing framework to the non-trivial problem of location synchronization. They provide relevant theoretical guarantees and demonstrate that their method achieves the lowest known sample complexity for exact recovery. Empirically, the approach appears very promising and represents a significant advancement in global location estimation.

**Limitations:**

Yes

**Quality:**

4

**Strengths And Weaknesses:**

Strengths:

1. The authors propose a comprehensive framework for robust camera translation estimation by:

     (a) Introducing a novel technique based on distance-based cycle-consistency, enabling accurate estimation of camera locations from input translation directions.

     (b) Applying robust modifications to each step that influences the accuracy of translation directions, including (i) a modified cycle-consistency based rotation averaging step, and (ii) the use of matched point coordinates to estimate robust relative translation directions via subspace recovery (STE).

2. The paper is generally clear, well-organized, and accessible to readers familiar with Structure-from-Motion pipelines.

3. Strong theoretical guarantees are provided, demonstrating that the method can accurately recover corruption levels of relative translations even under arbitrary adversarial corruption.

4. The experimental evaluation is extensive, with results reported on both synthetic and real datasets, across a variety of corruption models (uniform and adversarial). The method is compared both as a standalone location solver and within end-to-end SfM pipelines that include bundle adjustment, providing a comprehensive assessment of its practical value.

Weaknesses:
1. Theorem 2.1 lacks intuitive interpretation, particularly in terms of the values taken by the right-hand side (RHS) of the inequalities involving $s_{ij}^{(t)}$ for inlier and outlier edges. For the recovery guarantee to be meaningful, the RHS for inliers should ideally tend toward zero, while that for outliers should be significantly larger—indicating a clear separation between the two. However, the paper does not provide any insight into the typical magnitudes or range of these RHS terms, making it difficult to assess how strong or practical the theoretical guarantee really is. A small illustrative example or further discussion quantifying the behavior of the RHS terms would greatly improve the clarity and usefulness of this result.
2. The method relies on a carefully chosen initialization of weights $w_{ij}$ using $\tilde{s}_{ij}$ from T-AAB, but it’s unclear whether such meticulous design is truly necessary. If so, this should be justified through a sensitivity analysis showing how performance varies under alternative initializations.
3. The ablation studies are somewhat incomplete, as they do not evaluate the effect of removing or replacing individual components in the pipeline. For example, it remains unclear how a variant such as STE + LUD + MPLS-cycle would perform. Based on the results, it appears that a significant portion of the improvement may be attributable to the STE step rather than the proposed cycle-consistency refinement, which is positioned as the primary contribution. Additionally, it would be informative to evaluate the performance of BATA in place of LUD within the pipeline, especially since BATA is known to yield better solutions in scenarios with high variability in edge lengths.
4. In the comparison against SfM pipelines such as GloMap and Theia, the results shown in Figure 9 of the supplementary material (a corrected version of upper subfigure of Figure 4 in the main paper) indicate that Theia actually outperforms the proposed method on 9 out of 13 datasets. The average error metric, which favors the proposed method, appears to be disproportionately influenced by a few datasets where Theia performs poorly. Given the variability across datasets, the average may not be a reliable indicator of overall performance. In this context, the frequency of outperformance across datasets would be a more meaningful and fair comparison metric.
5. The performance of the proposed Cycle-sync method appears to depend on the specific annealing schedule for $\lambda_t$, which is currently fixed as $\frac{t}{t+10}$. It would be insightful to evaluate how the method behaves under different trajectories of $\lambda_t$. In particular, it would be useful to experiment with a schedule where $\lambda_t \rightarrow 0$ as $t$ increases—similar to MPLS—thereby placing more emphasis on cycle-consistency in early iterations and gradually shifting focus to the residuals. Additionally, analyzing the performance under a fixed $\lambda_t$ could help isolate the benefit of annealing itself. Such experiments would clarify the impact of the annealing strategy on the overall robustness and accuracy of the method.

---

> ### Author Rebuttal · Authors · 2025-07-31
>
> We are very grateful for your thoughtful and insightful comments. Below, we address each of your concerns in detail.
>
> **1. Intuitive interpretation of Theorem 2.1.**
>
> We appreciate the suggestion to provide more intuition. In Theorem 2.1, the separation between clean and corrupted edges arises from the fact that the upper bound for clean edges $\frac{1}{2\beta_0 r^t}$ vanishes as iteration $t\to \infty$ (note that $r>1$), while the lower bound for bad edges remains strictly positive, proportional to their corruption levels. This guarantees exact separation in the asymptotic regime. Although constants such as  $\mu$ and $C_\alpha$ can be intricate, they do not affect the eventual separation property. Moreover, our sample complexity result (Table 1) emphasizes the scaling order, which is more meaningful for understanding theoretical guarantees than specific constants.
>
> **2. Sensitivity to initialization**
>
> Our method is quite robust to initialization. While the T-AAB initialization accelerates convergence by providing better starting weights, it does not significantly influence the final accuracy. Even trivial initialization using uniform weights performs similarly after sufficient iterations. Below we report the mean and median location error on ETH3D data (averaged over different scenes) after several iterations for both initialization schemes:
>
> Trivial Initialization:
> | Iteration | mean error | median error |
> |:---------:|:--------:|:----------:|
> |   5     |    0.110    |   0.020  |
> |   10   |    0.100    |   0.017  |
> |   15   |    0.099    |   0.016  |
> |   20   |    0.099    |   0.016  |
>
>
> Our Initialization by T-AAB:
> | Iteration | mean error | median error |
> |:---------:|:--------:|:----------:|
> |   5     |    0.102    |   0.018  |
> |   10   |    0.099    |   0.016  |
> |   15   |    0.099    |   0.015  |
> |   20   |    0.099    |   0.014  |
>
> Therefore, our method is robust even to trivial initialization, let alone variations in the T-AAB parameter.
>
>
>
> **3 (a). The ablation studies STE + LUD + MPLS-cycle**
>
> This configuration is already included in the middle panel of Figure 4. As stated in lines 276–278, **all location solvers were preceded by the same STE** and MPLS-cycle preprocessing to ensure fairness. Without this common preprocessing, the performance gap between other baselines and Cycle-Sync would be even larger.
>
> **3 (b). Replacing LUD with BATA in our pipeline.**
>
> We appreciate the suggestion to try BATA within our reweighting framework. Yes, this is another advantage of cycle-sync, that it can incorporate any location solver in its pipeline, which is quite a general framework. Our experiments on ETH3D data show that integrating LUD under Cycle-Sync's reweighting leads to better performance compared to using integrating BATA under Cycle-Sync. It is likely due to the stronger constraint of LUD for anti-collapsing (the constraint is on every edge). Moreover, our Welsh-type objective function has already taken care of the large variation of distances, so angle based method (such as BATA) is not advantageous in this case.
>
> | Method | mean error | median error |
> |:---------:|:--------:|:----------:|
> |   Ours-LUD   |    0.099    |   0.014  |
> |   Ours-BATA   |    0.202    |   0.079  |
>
>
> **4. Frequency of Outperformance vs. Mean Error**
>
> Thanks a lot for your suggestion! We agree that frequency of outperforming baselines is a meaningful metric and will include it in our revision. While Theia performs better on 9 of 13 datasets, our method consistently avoids failure cases and achieves the lowest average error. We believe both metrics are informative: the former reflects robustness across diverse scenarios, while the latter captures stability and resilience against more adversarial datasets.
>
> **5. Annealing Schedule $\lambda_t$.**
>
> Thank you for this important observation. We have tested multiple schedules, including fixed, increasing and decreasing ones. Our proposed schedule strikes a good balance between residual-driven updates early on and cycle-consistency emphasis later. This schedule has consistently outperformed alternatives, particularly in settings with both additive noise and high corruption.
>
> Per the request of reviewer jMZZ, we have included experiments with both high corruption and noise. The following experiment uses uniform corruption with $q=0.7$ and higher noise level $\sigma=0.2$.
>
> | $\lambda_t$ | $\frac{t}{10+t}$ (ours) |  $0$ |  $\frac{10}{10+t}$|  $1$| $\frac{t}{t+5}$|
> |:-------------------------:|:--------:|:----------:|:----------:|:----------:|:----------:|
> |   median err  |  0.24      |    0.36 |  0.27 | 0.37 | 0.29|
>
>
> The following experiment test the adversarial corruption with $q=0.45$ (close to the theoretical limit $q=0.5$) and higher noise level $\sigma=0.2$.
>
>
> | $\lambda_t$ | $\frac{t}{10+t}$ (ours) |  $0$ |  $\frac{10}{10+t}$|  $1$| $\frac{t}{t+5}$|
> |:-------------------------:|:--------:|:----------:|:----------:|:----------:|:----------:|
> |   median err  |  0.17      |    0.32 |  0.18 | 0.20 | 0.17|
>
> We remark that using only the residual for reweighting (i.e., setting $\lambda_t = 0$, which corresponds to IRLS) often leads to significantly higher errors compared to cycle-based reweighting methods. The underlying issue is that, in noisy settings, some bad edges may coincidentally exhibit low residuals. When this occurs, the aggressive reweighting imposed by the Welsch objective can assign them disproportionately large weights, thereby amplifying their adverse impact. In contrast, our cycle-based reweighting effectively overcomes this limitation: it is extremely unlikely for a bad edge to exhibit a low average cycle inconsistency, unless all cycles it participates in are consistent—an event that is highly improbable for corrupted measurements. Overall, we find that our annealing strategy consistently achieves the best performance across most scenarios.
>
> We also observe this trend in the context of rotation synchronization. Supplementary Figure 10 illustrates that emphasizing cycle-consistency can significantly reduce orientation error. However, for rotation there is no need for annealing as it does not rely on distance estimation.
>
>
> **Q1: Prioritizing Residuals in Early Iterations**
>
> You are correct that the residual and cycle-inconsistency can both be unreliable in early iterations. However, cycle-based inconsistencies aggregate noise across three edges and are thus more vulnerable to early-stage corruption. As the iterations proceed and the distances become more accurate, the weighted average of cycle inconsistencies provides a more robust estimate of edge corruption. By aggregating information from neighboring edges, it reduces the likelihood of mistakenly identifying a bad edge as good.
>
>
> **Q2: why Welsch loss function?**
>
>
> While Geman-McClure and Huber losses also reduce the influence of large residuals, the Welsch loss flattens much faster due to its exponential decay, making it more effective at suppressing large errors.
>
> We further modify the Welsch loss by replacing $x^2$ with $|x|$ in the exponent, introducing a non-differentiable point at zero. This creates a singularity in the reweighting function, allowing good edges to receive infinite weight (capped in practice) and bad edges to be completely ignored—enabling exact recovery in corruption-only settings. In contrast, standard Geman-McClure, Huber, or Welsch losses always assign finite weights to good edges and nonzero weights to bad ones, preventing exact recovery.
>
> Although this weighting scheme appears aggressive, our cycle-based message passing effectively avoids misidentifying bad edges and mitigates the limitations of the objective function.
>
> **Q3: meaning of "disabling reweighting" in Lines 206-207?**
>
> This refers to disabling residual-based IRLS reweighting in rotation averaging. The weighting relies solely on cycle-consistency in MPLS-cycle. We will clarify this in the revised version.
>
> **Q4: Typo in Algorithm 1**
>
> Yes, thanks for pointing this out. We will correct it in the revised version.
>
> **Q5: Experiment on 1DSfM datasets?**
>
> In our earlier submission, we evaluated our method (we did not use STE and MPLS-cycle at the time) on the 1DSfM (photo tourism) datasets and achieved at least a 20% reduction in median error compared to the best baseline. However, reviewers raised concerns about our reliance on SIFT features and recommended switching to deep learning-based features. Another critique was that the ground truth camera poses in 1DSfM are generated by Bundler, which is considered outdated compared to modern tools like COLMAP—and even COLMAP may lack the accuracy of laser scans. This is particularly problematic when benchmarking high-precision location solvers. These limitations raise valid concerns about the suitability of older datasets for evaluating modern methods. Nevertheless, we report the mean and median location errors below (averaged over different scenes), with all methods consistently preprocessed using STE and MPLS-cycle:
>
> | Method | mean error | median error |
> |:-------------------------:|:--------:|:----------:|
> |   Ours   |   0.292    |   0.115  |
> |   LUD   |    0.314    |   0.132  |
> |   BATA   |    0.362    |   0.130  |
> |   ShapeFit   |    0.670    |   0.410  |
> |   FusedTA   |    0.330    |   0.130  |
>
> For the above reasons, we included an additional experiment in the supplementary material on a city-scale dataset (an extended version of Photo Tourism) using deep learning-based keypoint features (see Table 10 and Figure 11). However, the improvement of Cycle-Sync is much smaller than on ETH3D. We observed that LoFTR tends to overfit repetitive structures like windows and facades on this dataset. Furthermore, the ground truth camera poses are generated by COLMAP, which is likely less accurate than laser scans.

---

> > ### Comment · Reviewer_37Zk · 2025-08-02
> >
> > I thank the authors for their detailed rebuttal. The clarifications have addressed almost all of my queries adequately. I have only a couple of follow-up questions regarding the explanations provided:
> > Q2: It is encouraging that the proposed method performs well even with a trivial initialization. However, if the error–iteration trajectories for both the trivial and the T-AAB initializations are similar, what is the rationale for introducing the more elaborate T-AAB initialization?
> > Q5: Could you clarify whether the reported errors correspond to the 1DSfM datasets?

---

> > > ### Author Response · Authors · 2025-08-02
> > > **T-AAB Initialization and Results in Q5**
> > >
> > > Dear reviewer 37Zk,
> > >
> > > Thank you very much for carefully reading our rebuttal and for your prompt and thoughtful feedback!
> > >
> > > The T-AAB module is computationally efficient—it does not involve any least squares steps (which are typically the computational bottleneck) and runs in just a few seconds. In this sense, it is almost a "free lunch" that comes with theoretical guarantees (as provided in our paper). On the ETH3D dataset, T-AAB indeed yields slightly better results than trivial initialization within the same number of iterations. On other, simpler datasets, however, T-AAB can significantly reduce the overall runtime. For example, in the synthetic setting with uniform corruption and $q<0.5$, T-AAB perfectly separates good and bad edges (assigning an estimated corruption level of almost zero to good edges). As a result, with T-AAB weighting, Cycle-Sync (or any weighted least squares method) achieves exact recovery in just one iteration. In contrast, without T-AAB, Cycle-Sync on this dataset requires several more iterations of weighted least squares to converge, leading to a significantly longer runtime.
> > >
> > > Regarding the error table for Q5: it pertains to the 1DSfM dataset, where the ground truth camera poses are obtained from Bundler. To ensure fairness, all methods are preprocessed with STE and MPLS-cycle. Without this preprocessing, the reported errors for all methods would be substantially higher.

---

> ### Comment · Reviewer_37Zk · 2025-08-05
>
> I thank the authors for the clarification. I still have a few questions:
>
> 1. It remains unclear to me how Theorem 2.1 is specific to the use of the iteratively reweighted AAB algorithm. Since AAB is employed only for initialization, could the authors clarify whether the theorem's guarantees rely on this specific initialization? If so, how?
>
> 2. Additionally, the authors mention that only well-shaped triangles are used for computing $\tilde{s}_{ij}$.
>
>     Is this filtering applied solely during the computation of $\tilde{s}_{ij}$ or is it enforced throughout the estimation process? Clarifying this would help in understanding the robustness and applicability of the proposed approach. I think even the baseline methods would benefit tremendously if we do this filtering (Sensitivity in Translation Averaging, NeurIPS 2023).

---

> > ### Author Response · Authors · 2025-08-06
> > **Clarification about Theorem 2.1 (the first question)**
> >
> > Thank you for your thoughtful questions!
> >
> > **Regarding our first question:**
> >
> > We are happy to clarify the role and scope of Theorem 2.1 in our paper.
> >
> > This theorem provides a rigorous guarantee for the separation of clean and corrupted edges using cycle-inconsistency scores computed via a truncated AAB-style reweighting scheme (T-AAB). The analysis is specific to the initialization phase of our method, where we estimate corruption levels prior to running the main optimization (Algorithm 1). Under the setting of pure corruption without additive noise, exact recovery is information-theoretically possible. The theorem formally shows that the iteratively refined scores $\tilde{s}_{ij,t}$ converge to zero for clean edges and remain bounded away from zero for corrupted ones.
> >
> > This enables reliable identification of the good edge set, upon which any standard solver (including ours) can be applied to recover the ground-truth locations—either by discarding the bad edges or assigning them zero weight. In this way, Theorem 2.1 provides a provable path to exact recovery under ideal conditions.
> >
> > More broadly, the result highlights the power of cycle-consistency information for identifying outliers in camera location estimation. It also serves as a key theoretical motivation for the core design of Cycle-Sync, which integrates cycle-based message passing into the optimization.
> >
> > That said, as noted in our conclusion, this theoretical guarantee does not extend to the full nonconvex optimization pipeline. Extending such guarantees is challenging due to the interaction between nonconvexity, adaptive weights, and residual reweighting. Nevertheless, we view Theorem 2.1 as a meaningful first step, and extending the theory to cover the full pipeline remains an important and compelling direction for future work.
> >
> > We hope this clarifies the significance of Theorem 2.1 and its role in supporting both the design and theoretical foundation of our approach.

---

> > > ### Author Response · Authors · 2025-08-06
> > > **On the filtering of ill-shaped triangles (the second question)**
> > >
> > > **Regarding your second question**
> > >
> > > Thank you for the thoughtful question regarding our use of well-shaped triangles.
> > >
> > > To clarify, filtering ill-shaped triangles is applied only during the T-AAB initialization phase, when computing the corruption scores $\tilde{s}_{ij}$. It is not used at any point in the main Cycle-Sync optimization process, including the message-passing reweighting steps. This design choice ensures a fair comparison: all methods—including ours—operate on the same unfiltered input graph throughout the core estimation stage.
> > >
> > > Moreover, the triangle filtering in T-AAB does not remove any edges. If an edge does not participate in any well-shaped 3-cycle, we conservatively set its score $\tilde{s}_{ij} = 1$, indicating high corruption, but the edge remains in the graph. We will clarify this in the revision to avoid confusion.
> > >
> > > Our motivation for filtering triangles in the initialization phase is primarily theoretical. As shown in Theorem A.1, the stability constant $C_\alpha$ that governs the behavior of $\tilde{d}_{ij,k}$ can become arbitrarily large for ill-shaped triangles (e.g., those with extremely small or large angles). This instability prevents any meaningful theoretical separation of clean and corrupted edges. To rigorously prove the guarantee in Theorem 2.1, it is therefore essential to restrict attention to well-shaped cycles. The filtering is a theoretical necessity for exact recovery—not an algorithmic shortcut.
> > >
> > >
> > > In practice, we observe that filtering out badly shaped triangles is a simple and low-cost modification that improves the estimation of $\tilde{s}_{ij}$ and often accelerates convergence—particularly on synthetic datasets. Given its minimal computational overhead and consistent gains during initialization, it is a natural enhancement. However, the effect on final accuracy is minimal. Our method is robust even with trivial initializations, as we show in the rebuttal.
> > >
> > > We intentionally avoid triangle filtering in the main loop because it would reduce the size and connectivity of the graph, potentially eliminating useful constraints and affecting fairness in benchmarking. As shown in *Sensitivity in Translation Averaging* (NeurIPS 2023), such filtering can improve robustness but comes at the cost of operating on a smaller graph (see their Table 2). To maintain an apples-to-apples comparison, we preserve the full graph across all methods.
> > >
> > > While filtering ill-shaped triangles may offer some benefit to other baselines, we emphasize that triangle shape primarily affects the *stability* of the cycle-inconsistencies and residuals, rather than directly signaling corruption. In contrast, STE is explicitly designed to detect corrupted direction measurements and is therefore more aligned with the core objective of robust location estimation. For this reason, we apply STE uniformly across all baselines to ensure fair and more effective preprocessing (and indeed the improvements are significant), while reserving triangle filtering for the theoretical analysis in T-AAB, where stability is essential for proving exact recovery guarantees.

---

> > > > ### Comment · Reviewer_37Zk · 2025-08-06
> > > >
> > > > Thank you for the detailed responses. I think my questions are all answered satisfactorily.

---

### Note · Authors · 2025-08-15

We thank the reviewers for their time and feedback, which we have addressed carefully. This work offers an uncommon combination in structure-from-motion: a state-of-the-art robust location solver paired with a foundational theoretical advance.

Empirical Strength of Cycle-Sync:
Cycle-Sync consistently outperforms prior translation-averaging methods. As shown in Figure 4, it reduces the error of the best baseline by more than 60% on ETH3D, both with and without STE preprocessing. As a complete pipeline, it matches or exceeds the performance of Theia and GLOMAP. On synthetic data, even without STE, Cycle-Sync achieves up to five orders of magnitude (≈10^5×) lower error than competing solvers.

Theory:
We establish the strongest deterministic exact-recovery guarantee to date for location estimation under adversarial corruption. Under standard probabilistic models, our phase-transition bounds strictly improve upon those of ShapeFit and LUD (Table 1), whose results appeared in leading mathematical journals. To our knowledge, no work in the past seven years has advanced this specific theoretical frontier.

Datasets:
We use ETH3D as the primary benchmark because it provides centimeter-accurate laser-scan ground truth and calibrated imagery. For completeness, we also report IMC-PT results in the supplement and can include additional results as outlined in our reviewer response. The current experiments—both in the main paper and supplement—are already extensive and required substantial effort. We note that IMC-PT and 1DSfM-PT (addressed in our response to the reviewers) use proxy ground truth from SfM pipelines (COLMAP and Bundler, respectively), which can introduce circularity and make them less reliable for assessing absolute location accuracy.

Clarification:
Reviewer jMZZ initially raised seven highly technical issues, all of which we addressed. They later added two further concerns, drawn from other reviewers’ comments, which we had already addressed. Assuming no adversarial intent, we believe these reflect a misunderstanding: (1) STE is modular and optional, included only to demonstrate practical compatibility; Cycle-Sync’s core gains hold with or without it. NeurIPS policy should not discourage reporting such complementary, practitioner-relevant results that build on prior work. (2) Our theoretical results strengthen the best-known phase-transition guarantees for exact recovery in location estimation, advancing a benchmark that has stood for seven years.

---

### Decision · Program_Chairs · 2025-09-17

**Decision:**

Accept (spotlight)

**Comment:**

The paper received overwhelming positive reviews, three accepts and one borderline accept. AC sees now reason to diverge from the reviewers' recommendation.